

# Understanding the catalytic role of oxalic acid in the SO$_3$ hydration to form H$_2$SO$_4$ in the atmosphere

Guochun Lv[1], Xiaomin Sun[1,*], Chenxi Zhang[2], Mei Li[3,4*]

[1]Environment Research Institute, Shandong University, Jinan 250100, China

[2]College of Biological and Environmental Engineering, Binzhou University, Binzhou 256600, China

[3]Institute of Mass Spectrometer and Atmospheric Environment, Jinan University, Guangzhou 510632, China

[4]Guangdong Provincial Engineering Research Center for on-line source apportionment system of air pollution, Guangzhou

510632, China

[*]*Corresponding authors*: Xiaomin Sun (sxmwch@sdu.edu.cn); Mei Li (limei2007@163.com)

**Abstract:** The hydration of SO$_3$ plays an important role in atmospheric sulfuric acid formation. Some atmospheric species can involve in and facilitate the reaction. In this work, using quantum chemical calculations, we show that oxalic acid, the most common dicarboxylic acid in the atmosphere, can effectively catalyze the hydration of SO$_3$. The energy barrier of SO$_3$

hydration reaction catalyzed by oxalic acid (cTt, tTt, tCt and cCt conformers) is about or below 1 kcal mol$^{-1}$, which is lower than the energy barrier of 5.73 kcal mol$^{-1}$ for water-catalyzed SO$_3$ hydration. By comparing the rate of SO$_3$ hydration reaction catalyzed by oxalic acid and water, it can be found that the oxalic acid-catalyzed SO$_3$ hydration can compete with water-catalyzed SO$_3$ hydration in the upper troposphere. This leads us to conclude that the involvement of oxalic acid in SO$_3$ hydration to form H$_2$SO$_4$ is significant in the atmosphere.

## 1. Introduction

In the atmosphere, hydrogen atom transfer (HAT) reactions play a significant role in many processes. The radical reaction (R-H+OH→R+H$_2$O), as the most traditional HAT reactions, can be widely found in the atmosphere (Alvarez-Idaboy



et al., 2001;Cameron et al., 2002;Steckler et al., 1997;Atkinson et al., 2006). Some addition (Steudel, 1995;Williams et al., 1983;Courmier et al., 2005;Zhang and Zhang, 2002), decomposition (Rayez et al., 2002;Kumar and Francisco, 2015;Gutbrod et al., 1996), isomerization (Zheng and Truhlar, 2010;Atkinson, 2007), and abstraction reactions (Ji et al., 2013;Ji et al., 2017) also are the important HAT reaction in the atmosphere. These atmospheric HAT reactions have a main feature that two-point

hydrogen bond can occur and can facilitate hydrogen atom transfer (Kumar et al., 2016). The presence of water, acid and other catalysts, as the hydrogen donors and acceptors, contributes to the formation of two-point hydrogen bond (Vöhringer-Martinez et al., 2007;da Silva, 2010;Gonzalez et al., 2010;Bandyopadhyay et al., 2017). Thus, the effect of catalysts on promoting atmospheric HAT reactions has attracted more attention of atmospheric scientists.

The hydration of $SO_3$ to form sulfuric acid is a typical addition reaction involving the hydrogen atom transfer. In the

atmosphere, this hydration reaction is regarded as the main source of gas-phase sulfuric acid. For the reaction $SO_3+H_2O \rightarrow H_2SO_4$, the pre-reactive $SO_3 \cdots H_2O$ complex firstly is formed, and the complex then rearrange to form $H_2SO_4$, which was proposed by Castleman et al (Holland and Castleman, 1978;Hofmann-Sievert and Castleman, 1984). But the subsequent research found that, due to high energy barrier, this hydration reaction involving a single water molecule cannot takes place in the atmosphere (Hofmann and Schleyer, 1994;Morokuma and Muguruma, 1994;Steudel, 1995). The inclusion

of a second water molecule in the above reaction has been proven to significantly reduce the hydration energy barrier (Morokuma and Muguruma, 1994;Loerting and Liedl, 2000;Larson et al., 2000). The promoting effect can be mainly attributed to the formation of the two-point hydrogen bond, which reduce the ring strain occurring in the pre-reactive complex, and facilitates the rearrangement of the pre-reactive complex via double hydrogen atom transfer. It has also been shown that some other atmosphere molecules can behave as a catalyst to promote the hydration of $SO_3$. To the best of our

knowledge, hydroperoxy radical (Gonzalez et al., 2010), formic acid (Hazra and Sinha, 2011;Long et al., 2012), sulfuric acid (Torrent-Sucarrat et al., 2012), nitric acid (Long et al., 2013) and ammonia (Bandyopadhyay et al., 2017) have been reported to replace the second water to catalyze the hydration reaction of $SO_3$.

Oxalic acid (OA), the most prevalent dicarboxylic acid in the atmosphere (Ho et al., 2015;Kawamura and Ikushima,



1993), is the water-soluble organic acid, so it has high concentration in aerosols (Kawamura et al., 2013;van Pinxteren et al., 2014;Deshmukh et al., 2016;Wang et al., 2016). In addition to accumulating in aerosols, oxalic acid, as an organic acid in the gas phase, has been found to enhance the new particle formation (NPF) (Xu et al., 2010;Weber et al., 2012;Xu and Zhang, 2012;Weber et al., 2014;Peng et al., 2015;Miao et al., 2015;Zhao et al., 2016;Chen et al., 2017;Xu et al., 2017;Arquero et al.,

2017;Zhang, 2010). Theoretical studies about the effect of oxalic acid on atmospheric particle nucleation and growth have shown that it can form stable complexes with water (Weber et al., 2012), sulfuric acid (Xu et al., 2010;Xu and Zhang, 2012;Miao et al., 2015;Zhao et al., 2016), ammonia (Weber et al., 2014;Peng et al., 2015) and amines (Chen et al., 2017;Xu et al., 2017;Arquero et al., 2017) via intermolecular hydrogen bond. The potential of oxalic acid for contributing to the NPF is mainly attributed to its capability of forming hydrogen bond with hydroxyl and/or carbonyl-type functional group. Thus, it

can be believed that oxalic acid is good candidate for catalyzing the hydrogen atom transfer reaction in the atmosphere.

In this paper, we reported the hydration reaction of $SO_3$ in the presence of oxalic acid, aiming to study the catalytic effect and importance of oxalic acid in the hydration of $SO_3$. As is known, oxalic acid can exist in several conformational forms (Buemi, 2009), which can be identified through the nomenclature used by Niemen et al (Nieminen et al., 1992). Thus, five stable conformers of oxalic acid were considered in this work. The rate constants of oxalic acid-catalyzed $SO_3$ hydration

were calculated using the kinetics analysis, and compared with that for water-catalyzed hydration reaction. Finally, combining concentrations of reactants with the rate constants, we evaluated the importance of the hydration process involving the oxalic acid relative to the hydration of $SO_3$ in the second water as a catalyst to form sulfuric acid.

**2. Computational details**

Gaussian 09 suit of software (Frisch et al., 2010) were used in this work to perform all electronic structure calculations.

The geometric structures including all reactant, complex, transition state and products were optimized using M06-2X method (Zhao and Truhlar, 2008) with 6-311++G(3df,3pd) basis set. For M06-2X method, the ultrafine integration grid was chosen to enhance calculation accuracy at reasonable additional cost. The frequency calculations were carried out with the same level after geometric optimization to verify the local minimum points and transition states through the criterion that no



imaginary frequencies for the local minimum point and one imaginary frequency for transition states. According to

frequency calculations, the zero point energies (ZPE) and thermal corrections also can be obtained. The intrinsic reaction

coordinate (IRC) calculation (Fukui, 1981;Hratchian and Schlegel, 2004;Hratchian and Schlegel, 2005) was performed to

ensure that the transition states connected with the corresponding reactants and products. A high level ab initio method,

CCSD(T) method (Purvis and Bartlett, 1982;Pople et al., 1987), with the 6-311++G(3df,3pd) basis set was used to refine the

single-point energies of these optimized species. To obtain the conformational population of oxalic acid in different

temperature more accurately, the quantum chemistry composite method, Gaussian 4 (G4) theory (Curtiss et al., 2007), also

was performed for oxalic acid conformers.

For the kinetics analysis, the electronic energies based on the CCSD(T)/6-311++G(3df,3pd) level of theory, while the

partition functions obtained from the M06-2X/6-311++G(3df,3pd) level of theory. The rate constants for the rearrangement

process of $SO_3$ hydration reaction was estimated using conventional transition-state theory (TST) (Truhlar et al., 1996) with

Wigner tunneling correction. All kinetics analysis was executed in the KiSThelP program (Canneaux et al., 2014). The

kinetics analysis is summarized as follows.

On the basis of the discussion in this paper, it can conclude that the hydration reactions begin with the formation of

pre-reactive complex, and then undergo a transition state to form post-reactive complex. This process can be characterized

by the following equation:

$$A + B \underset{k_{-1}}{\overset{k_1}{\rightleftharpoons}} pre-reactive\ complex \xrightarrow{k_{uni}} post-reactive\ complex \qquad (1)$$

Assuming that the pre-reactive complex is in equilibrium with the reactants and the steady state approximation is

applied to pre-reactive complex, the reaction rate can be formulated as

$$v = \frac{k_1}{k_{-1}} k_{uni}[A][B] = K_{eq} k_{uni}[A][B] \qquad (2)$$

where $K_{eq}$ is the equilibrium constant of the first step and $k_{uni}$ is the rate constant for unimolecular reaction of pre-reactive

complex to post-reactive complex.





For water-catalyzed hydration process, its two channels can be written as:

*Recation* 1:

$$SO_3 \cdots H_2O + H_2O \underset{k_{-1}}{\overset{k_1}{\rightleftharpoons}} SO_3 \cdots H_2O \cdots H_2O \xrightarrow{k_{uni\_w}} H_2SO_4 \cdots H_2O \quad (3)$$

*Reaction* 2:

$$SO_3 + H_2O \cdots H_2O \underset{k_{-2}}{\overset{k_2}{\rightleftharpoons}} SO_3 \cdots H_2O \cdots H_2O \xrightarrow{k_{uni\_w}} H_2SO_4 \cdots H_2O \quad (4)$$

The corresponding rate constants are that:

$$v_{SO_3 \cdots H_2O + H_2O} = \frac{k_1}{k_{-1}} k_{uni\_w}[SO_3 \cdots H_2O][H_2O] = K_{eq1} k_{uni\_w}[SO_3 \cdots H_2O][H_2O] \quad (5)$$
$$= k_{w1}[SO_3 \cdots H_2O][H_2O]$$

$$v_{SO_3 + H_2O \cdots H_2O} = \frac{k_{w2}}{k_{-w2}} k_{uni\_w}[SO_3][H_2O \cdots H_2O] = K_{eq2} k_{uni\_w}[SO_3][H_2O \cdots H_2O] \quad (6)$$
$$= k_{w2}[SO_3][H_2O \cdots H_2O]$$

For oxalic acid-catalyzed hydration reaction of $SO_3$, it also has two reaction channels and has the similar features as the water-assisted hydration process. The two channels can be shown as following:

*Reaction* X1:

$$SO_3 \cdots H_2O + X \underset{k_{-X1}}{\overset{k_{X1}}{\rightleftharpoons}} SO_3 \cdots H_2O \cdots X \xrightarrow{k_{uni\_X}} H_2SO_4 \cdots X' \quad (7)$$

*Reaction* X2:

$$SO_3 + H_2O \cdots X \underset{k_{-X2}}{\overset{k_{X2}}{\rightleftharpoons}} SO_3 \cdots H_2O \cdots X \xrightarrow{k_{uni\_X}} H_2SO_4 \cdots X' \quad (8)$$

Here, the symbol X in these two equations represents the different conformers of oxalic acid (namely, cTt, tTt, tCt and cCt conformer).

The corresponding rate of $SO_3$ hydration involving in oxalic acid can be obtained as following:

$$v_{SO_3 \cdots H_2O + X} = \frac{k_{X1}}{k_{-X1}} k_{uni\_X}[SO_3 \cdots H_2O][X] = K_{eq\_X1} k_{uni\_X}[SO_3 \cdots H_2O][X] \quad (9)$$
$$= k_{X1}[SO_3 \cdots H_2O][X]$$





$$v_{SO_3+H_2O\cdots X} = \frac{k_{X2}}{k_{-X2}}k_{uni\_X}[SO_3][H_2O\cdots X] = K_{eq2\_X2}k_{uni\_X}[SO_3][H_2O\cdots X]$$
$$= k_{X2}[SO_3][H_2O\cdots X]$$
(10)

To assess the importance of oxalic acid in $SO_3$ hydration to $H_2SO_4$ in the atmosphere, the relative rate can be used as

$$\frac{v_{SO_3\cdots H_2O+X}}{v_{SO_3\cdots H_2O+H_2O}} = \frac{k_{X1}[SO_3\cdots H_2O][X]}{k_{w1}[SO_3\cdots H_2O][H_2O]} = \frac{k_{X1}[X]}{k_{w1}[H2O]}$$
(11)

$$\frac{v_{SO_3+H_2O\cdots X}}{v_{SO_3+H_2O\cdots H_2O}} = \frac{k_{X2}[SO_3][H_2O\cdots X]}{k_{w2}[SO_3][H_2O\cdots H_2O]} = \frac{k_{X2}[H_2O\cdots X]}{k_{w2}[H_2O\cdots H_2O]}$$
(12)

## 3. Results and discussion

### 3.1. Water-catalyzed hydration reaction of $SO_3$

Although the hydration of $SO_3$ involving two water molecules has been talked about many times, we still include it in our paper so as to compare this reaction with the following $SO_3$ hydration reaction catalyzed by oxalic acid at the same theoretical level, that is, at CCSD(T)/6-311++G(3df,3pd)//M06-2X/6-311++G(3df,3pd) level. For the reaction $SO_3+2H_2O$, the existence of two pathways has become a consensus. The one is that water dimer react with $SO_3$ to obtain pre-reactive complex, then this complex rearrange to form $H_2SO_4\cdots H_2O$ complex (channel 1); the other begin with the reaction of $SO_3\cdots H_2O$ complex with water, the following rearrangement is the same as the channel 1 (channel 2). The potential energy profile and geometric structures can be found from Fig. 1. Other results calculated about the reaction are put into Supplement (Table S1).

For the two channel, it is clear that the binding energy of water dimer is 3.22 kcal mol$^{-1}$, and that of $SO_3\cdots H_2O$ complex is 6.48 kcal·mol$^{-1}$. As shown in the Fig. 1, the two pathways share the same pre-reactive complex (RC1) and the following processes. The RC1 has the binding energy of 15.47 kcal mol$^{-1}$ relative to $SO_3+2H_2O$, and consist of six-membered ring structure, in which two hydrogen bonds (or called two-point hydrogen bond) between $H_a$ and $O_2$, $H_b$ and $O_3$, can be found. The formation of sulfuric acid from the RC1 needs to go through the rearrangement process with a transition state, which is the rate-limiting step with the barrier energy of 5.73 kcal mol$^{-1}$ with respect to the RC1. The post-reactive complex (PC1)



which is the complex of sulfuric acid with water is 13.34 kcal mol$^{-1}$ below the RC1. In addition, the binding energy of PC1 is

11.02 kcal mol$^{-1}$ compared to isolated sulfuric and water.

### 3.2. Oxalic acid-catalyzed hydration reaction of SO$_3$

Oxalic acid conformers, as shown in Fig. 2, are named according to the configurations of H-O-C-C and O=C-C=O

dihedral angle: the low-case letters refer to cis (c) or trans (t) configuration of H-O-C-C; the upper-case letters relate to cis (C)

or tans (T) configuration of O=C-C=O. The nomenclature about conformers of oxalic acid is proposed by Nieminen et

al(Nieminen et al., 1992).

The calculated potential energy profile for oxalic acid-catalyzed hydration reaction of SO$_3$ is depicted in Fig. 3, and the

corresponding geometry structures are shown in Fig. 4. Energies, enthalpies and free energies of all relevant species for

oxalic acid catalyzed hydration of SO$_3$ are summarized in Supplement (Table S2 - S5). From the two figures, it is obvious

that the cTc conformer cannot act as a catalyst in the hydration reaction because the hydrogen atom transfer cannot occur.

The failure of this transfer is due to that the hydrogen and oxygen atom involving in two-point hydrogen bond do not come

from the same carboxyl group (see Fig. 4). The hydration reactions catalyzed by the remaining four conformers have the

same feature (Fig. 3), which is that the pre-reactive complex formed from SO$_3$···H$_2$O complex with oxalic acid (OA) or

OA···H$_2$O complex with water can evolve into product complex (H$_2$SO$_4$···OA complex) via transition state of hydrogen

atom transfer. For the reaction catalyzed by cTt conformer, the binding energy of cTt···H$_2$O complex is 9.05 kcal mol$^{-1}$.

Compared to the cTt···H$_2$O complex with SO$_3$, the binding energy of pre-reactive complex is 11.24 kcal mol$^{-1}$, while that is

13.81 kcal mol$^{-1}$ relative to SO$_3$···H$_2$O complex with cTt. The transformation from RC$_{cTt}$ to PC$_{cTt}$ corresponds to hydrogen

atom transfer process, and has a transition state (TS$_{cTt}$) with the energy barrier of 1.37 kcal mol$^{-1}$ with respect to RC$_{cTt}$. The

post-reactive complex (PC$_{cTt}$) lies below the RC$_{cTt}$ by 13.11 kcal mol$^{-1}$. It should be noted that the cTt conformer from RC$_{cTt}$

has transform to the cCt conformer in PC$_{cTt}$. The binding energy of PC$_{cTt}$ is 18.54 kcal mol$^{-1}$ compared to H$_2$SO$_4$ and cCt

conformer.

For hydration reaction involving the tTt conformer, the tTt···H$_2$O complex is stabilized by 8.76 kcal mol$^{-1}$, relative to



tTt + $H_2O$. Starting with the tTt···$H_2O$ + $SO_3$ channel, the pre-reactive complex ($RC_{tTt}$) can be formed with binding energy of 12.82 kcal mol$^{-1}$, whereas $RC_{tTt}$ has the binding energy of 15.10 kcal mol$^{-1}$ when it come from the $SO_3$···$H_2O$ + tTt channel. The $TS_{tTt}$ lies above the $RC_{tTt}$ by 0.44 kcal mol$^{-1}$, and proceed with the formation of $PC_{tTt}$, which is 13.75 kcal mol$^{-1}$ more stable than $RC_{tTt}$. The formed $PC_{tTt}$ needs to overcome 18.06 kcal mol$^{-1}$ to disaggregate into $H_2SO_4$ and tCt conformer.

For tCt conformer, the binding energy of tCt···$H_2O$ complex is 8.93 kcal mol$^{-1}$. In the two channels starting from the tCt···$H_2O$ + $SO_3$ entry and $SO_3$···$H_2O$ + tCt entry, the same pre-reactive complex ($RC_{tCt}$) can be formed with the binding energy of 13.11 kcal mol$^{-1}$, 15.56 kcal mol$^{-1}$, respectively, with respect to the two reactants. The $RC_{tCt}$ proceeds via the transition state ($TS_{tCt}$) lying above $RC_{tCt}$ by 0.28 kcal mol$^{-1}$ into post-reactive complex ($PC_{tCt}$), which is 14.01 kcal mol$^{-1}$ more stable than the $RC_{tCt}$ complex. The $PC_{tCt}$ complex also can be formed from the $H_2SO_4$ and tTt conformer with the energy release of 17.73 kcal mol$^{-1}$.

For cCt conformer, the binding energy of cCt···$H_2O$ complex is 9.92 kcal mol$^{-1}$. Beginning with cCt···$H_2O$ + $SO_3$ channel and $SO_3$···$H_2O$ + cCt channel, the hydration reaction undergo the same $RC_{cCt}$ complex, which is more stable than the two reactants (by 12.61 kcal mol$^{-1}$, 16.05 kcal mol$^{-1}$, respectively), and a transition state ($TS_{cCt}$) with the energy barrier of 0.26 kcal mol$^{-1}$, to form the $H_2SO_4$···cTt complex ($PC_{cCt}$), which lies below $RC_{cCt}$ complex by 15.07 kcal mol$^{-1}$ and below $H_2SO_4$ + cTt by 16.87 kcal mol$^{-1}$.

In the light of the analysis above, it is obvious that energy barrier of the hydration reaction catalyzed by oxalic acid is about or below 1 kcal mol$^{-1}$. The result shows that the oxalic acid is more effective than water in catalyzing the $SO_3$ hydration because the hydration reaction catalyzed by water has energy barrier of nearly 6 kcal mol$^{-1}$. Another thing we want to mention is that one oxalic acid conformer involving in $SO_3$ hydration transfer to another conformer type after the completion of hydration reaction (cTt → cCt, tTt → tCt, tCt → tTt, cCt → cTt).

### 3.3. Atmospheric implications

For evaluating the importance of oxalic acid in enhancing $H_2SO_4$ formation in atmosphere, we calculated the rate constants of $SO_3$ hydration catalyzed by oxalic acid and water, and compared the rate of oxalic acid-catalyzed hydration



reaction with that of water-catalyzed hydration at different altitudes in the troposphere. The corresponding temperature, pressure, density of air and water vapor content at an altitude are taken from U.S. Standard Atmosphere, 1976 (NASA and NOAA), and are put into the Table S6.

According to these calculation methods described in computational details section, we can confirm that the ratio of rate

constant and relative concentrations, are both important elements to estimate the effect of oxalic acid-catalyzed $SO_3$ hydration. Thus, the rate constants for hydration reaction of $SO_3$ catalyzed by oxalic acid and water at different altitudes are firstly analyzed, and summarized in Table 1. The corresponding $K_{eq}$ and $k_{uni}$ are included in Supplement (Table S7).

As shown in Table 1, the rate constants for the Reaction cTt1 at 298.15 K and altitude of 0 km ($5.87 \times 10^{-8}$ cm$^3$ molecule$^{-1}$ s$^{-1}$) is about $10^4$ times than, whereas the Reaction tTt1, tCt1 and cCt1 ($k_{tTt1}$: $1.99 \times 10^{-6}$, $k_{tCt1}$: $3.10 \times 10^{-6}$, $k_{cCt1}$:

$1.02 \times 10^{-5}$ cm$^3$ molecule$^{-1}$ s$^{-1}$) are 6 orders of magnitude greater than, that for the Reaction 1 ($1.52\times10^{-12}$ cm$^3$ molecule$^{-1}$ s$^{-1}$). From the comparison of rate constants between the reaction of the $H_2O\cdots X$ complex with $SO_3$ (Reaction X2) and the reaction of water dimer with $SO_3$ (Reaction 2), it is also clear that the rate constant of Reaction X2 is higher than Reaction 2 (at 298.15 K and altitude of 0 km, $k_{cTt2}/k_{w2}$: $\sim10^2$, $k_{tTt2}/k_{w2}$: $\sim10^4$, $k_{tCt2}/k_{w2}$: $\sim10^4$, $k_{cCt2}/k_{w2}$: $\sim10^4$). All corresponding rate constant ratios at various altitudes are described in Table 2. It can be seen from Table 2 that rate constant ratios increase with

the temperature decreasing (that is, with altitude increasing).

Another factor in the rate comparison, as shown in Eq. (11) and Eq. (12), is the reactant concentration. For Eq. (11), the concentration considered is about oxalic acid conformers and water vapor, while the concentration in Eq. (12) is about oxalic acid-water complex and water-water complex. For simplicity, we will only talk about the rate comparison between the Reaction X1 and the Reaction 1 in the following part.

Based on the calculated Gibbs free energy at G4 level (see Table S8), and assuming a Boltzmann distribution, the mole fractions for oxalic acid conformers can be obtained (Table 3). The calculation method about the conformational population is shown in Supplement (Text S1). The most stable conformer, cTc conformer, accounts for more than 95 % of oxalic acid in the altitude range from 0 km to 12 km. But this conformer cannot participate in catalyzing the hydration reaction of $SO_3$. In

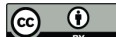

other words, the conformers involving in assisting the $SO_3$ hydration do not excess 5 % of oxalic acid. In some studies (Martinelango et al., 2007;Bao et al., 2012), observed concentration of oxalic acid in gas phase range from approximately 10 ng m$^{-3}$ to close 1 μg m$^{-3}$. To obtain actual concentrations of oxalic acid conformers, we assumed a 302 ng m$^{-3}$ ($2.02 \times 10^9$ molecule cm$^{-3}$) for total oxalic acid at 298.15 K and altitude of 0 km, which are measured by Bao et al (2012). Combining

mole fractions for conformers (cTt, tTt, tCt and cCt) and the total concentration of oxalic acid considering the temperature and pressure effect using idea gas equation (see Table S9), the concentration of oxalic acid conformers involving in the hydration reaction at various altitudes can be calculated, which is tabulated in Table 4. The water vapor concentration also is included in Table 4.

As shown in Table 5, the ratio of the total reaction rate for $SO_3 \cdots H_2O$ complex with oxalic acid conformers (cTt, tTt,

tCt and cCt) to the rate for $SO_3 \cdots H_2O$ complex with $H_2O$ are described. At altitude of 0 km, the rate ratio for these two reactions is in the range of $10^{-4}$-$10^{-5}$ at two temperatures (298.15 K and 288.15 K), which indicates that the oxalic acid-catalyzed $SO_3$ hydration cannot compete with the water-catalyzed hydration at 0 km with different temperatures. However, as the altitude increase, the oxalic acid has an increasing impact on the $SO_3$ hydration because of obvious increase of the ratio. When the altitude increases to 6 km, the water-catalyzed hydration reaction is approximately 55 times faster than

the oxalic acid-catalyzed $SO_3$ hydration. The oxalic acid-catalyzed reaction just is 1 order of magnitude slow than water-catalyzed reaction at altitude of 8 km. As the altitude rise to 10 km, the oxalic acid-catalyzed $SO_3$ hydration is competitive with water-assisted hydration. At altitude of 12 km, the oxalic acid-catalyzed hydration reaction is about 6 times faster than the water-assisted hydration.

To sum up, the comparison of relative rate for oxalic acid and water catalyzed $SO_3$ hydration reaction shows that the

oxalic acid can play a significant role in $SO_3$ hydration to $H_2SO_4$ in the upper troposphere.

## 4. Conclusion

The main conclusion of this work is that oxalic acid, the most abundant dicarboxylic acid in the atmosphere, has the remarkable ability to catalyze $SO_3$ hydration to $H_2SO_4$, and has the real impact on the $H_2SO_4$ formation in the atmosphere.

We have shown that water catalyzed hydration reaction of $SO_3$ has energy barrier of 5.73 kcal mol$^{-1}$. For oxalic acid,

four conformers (cTt, tTt, tCt and cCt) can involve in the hydration reaction, whereas the most stable conformer (cTc) cannot

catalyze the hydration reaction because the formed hydrogen bond structure rejects hydrogen atom transfer. The energy

barrier of hydration reaction of $SO_3$ is about or below 1 kcal mol$^{-1}$. The results signify oxalic acid has the higher catalytic

ability than water for $SO_3$ hydration to form $H_2SO_4$.

According to the kinetics analysis, the rate constant of oxalic acid-catalyzed hydration reaction is greater than that of

water-catalyzed reaction by orders of magnitude, which also reflects the obvious catalytic ability of oxalic acid. In addition

to the rate constant, the reactant concentration also is an important factor to assess the effect of oxalic acid on $SO_3$ hydration.

Based on the two factors, our calculation shows that the oxalic acid-catalyzed hydration reaction can compete with the

water-catalyzed reaction in the upper troposphere. The results indicate that, in the upper troposphere, the oxalic acid can play

an important role in $SO_3$ hydration to form $H_2SO_4$ in the atmosphere.

This work not only gives insight into the new mechanism of $SO_3$ hydration in the atmosphere, but also has potential

importance for investigating the catalytic effect of oxalic acid on other atmosphere reaction.

**Supplement**

**Competing interests**

The authors declare that they have no conflict of interest.

**Acknowledgments**

This work is supported by National Natural Science Foundation of China (21337001, 21577021 and 21607056), the

National Key Technology R&D Program (grant no.2014BAC21B01), the NSFC of Guangdong Province

(2015A030313339),and the Fundamental Research Funds for the Central Universities (21617455).

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

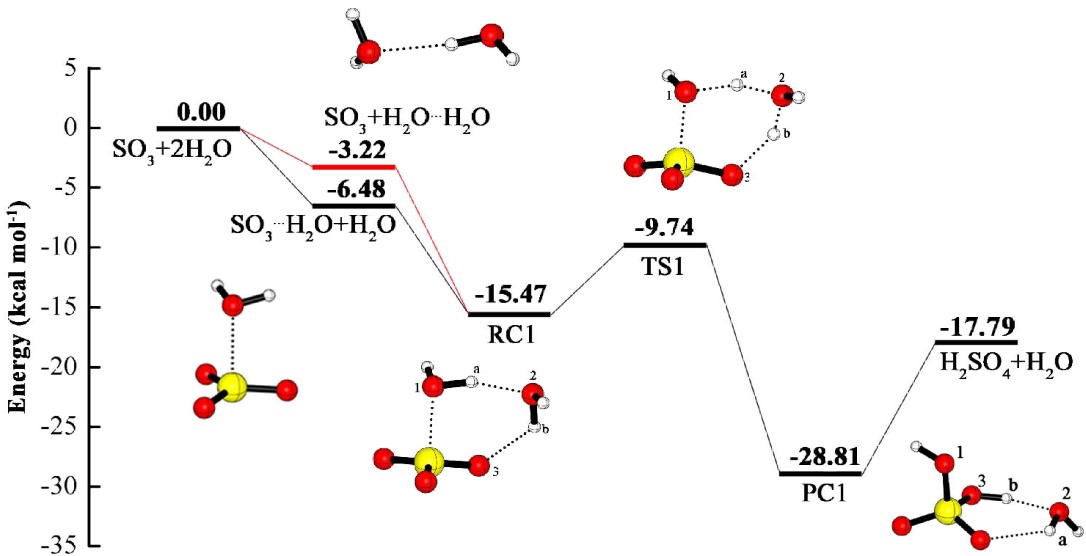

**Figure 1.** Calculated potential energy profile for the hydration of $SO_3$ with the second water as a catalyst at the CCSD(T)/6-311++G(3df,3pd)//M06-2X/6-311++G(3df,3pd) level.





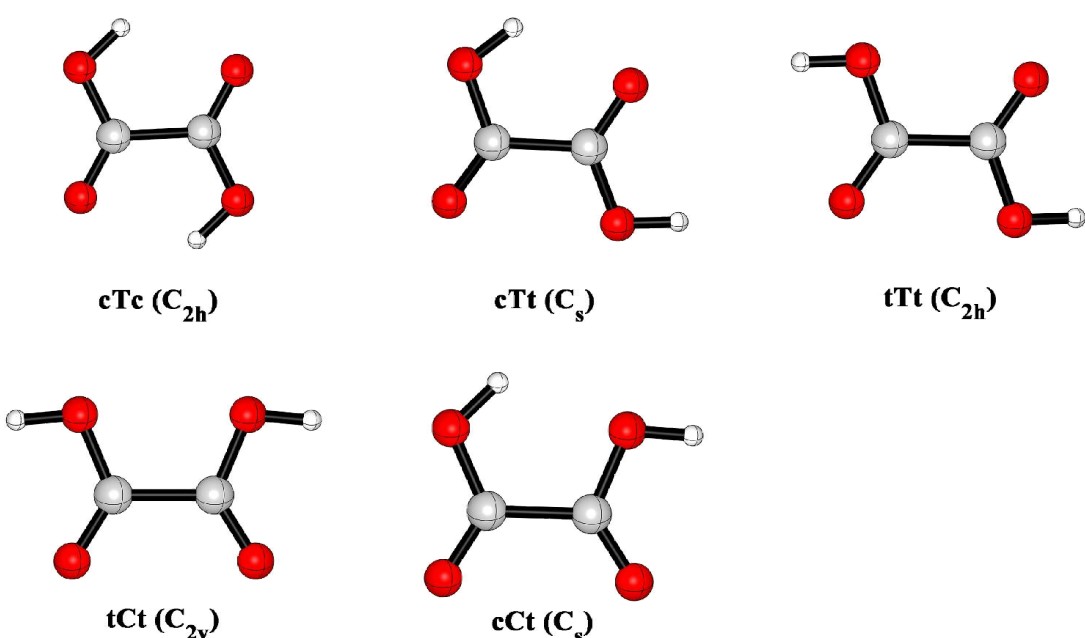

**Figure 2.** Structures of oxalic acid conformers optimized at the M06-2X/6-311++G(3df,3pd) level.





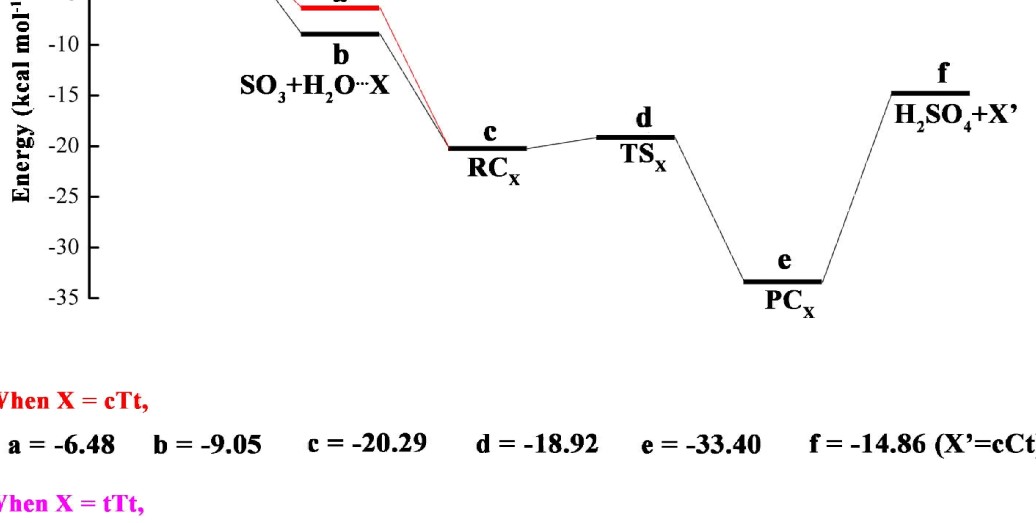

**When X = cTt,**

a = -6.48    b = -9.05    c = -20.29    d = -18.92    e = -33.40    f = -14.86 (X'=cCt)

**When X = tTt,**

a = -6.48    b = -8.76    c = -21.58    d = -21.14    e = -35.33    f = -17.27 (X'=tCt)

**When X = tCt,**

a = -6.48    b = -8.93    c = -22.04    d = -21.76    e = -36.05    f = -18.32 (X'=tTt)

**When X = cCt,**

a = -6.48    b = -9.92    c = -22.53    d = -22.27    e = -37.60    f = -20.73 (X'=cTt)

**Figure 3.** Calculated potential energy profile for the hydration of $SO_3$ with oxalic acid conformers (cTt, tTt, tCt and cCt) as catalysts at the CCSD(T)/6-311++G(3df,3pd)//M06-2X/6-311++G(3df,3pd) level.




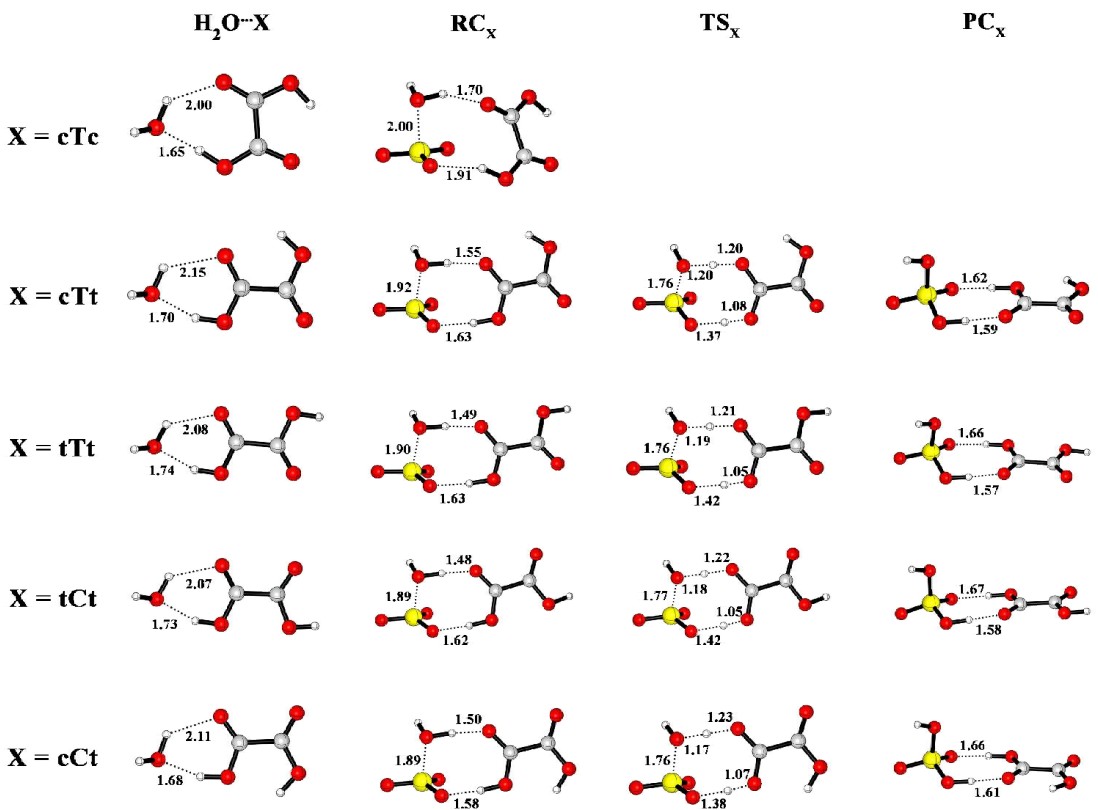

**Figure 4.** Optimized M06-2X/6-311++G(3df,3pd) structures of reactant complexes, pre-reactive complexes, transition states and post-reactive complexes for the oxalic acid-catalyzed $SO_3$ hydration reaction.



**Table 1.** Rate constants (in $cm^3$ molecule$^{-1}$ s$^{-1}$) of $SO_3$ hydration reaction catalyzed by water and by oxalic acid at different altitudes.

| altitude (km) | 0 | 0 | 2 | 4 | 6 | 8 | 10 | 12 |
|---|---|---|---|---|---|---|---|---|
| P (bar) | 1.01325 | 1.01325 | 0.795 | 0.617 | 0.472 | 0.357 | 0.265 | 0.194 |
| T (K) | 298.15 | 288.15 | 275.15 | 262.17 | 249.19 | 236.22 | 223.25 | 216.65 |
| $k_{w1}$ | $1.51\times10^{-12}$ | $2.01\times10^{-12}$ | $2.99\times10^{-12}$ | $4.62\times10^{-12}$ | $7.44\times10^{-12}$ | $1.27\times10^{-11}$ | $2.26\times10^{-11}$ | $3.13\times10^{-11}$ |
| $k_{w2}$ | $3.12\times10^{-12}$ | $5.06\times10^{-12}$ | $9.95\times10^{-12}$ | $2.08\times10^{-11}$ | $4.71\times10^{-11}$ | $1.16\times10^{-10}$ | $3.15\times10^{-10}$ | $5.46\times10^{-10}$ |
| $k_{cTt1}$ | $5.87\times10^{-8}$ | $1.26\times10^{-7}$ | $3.67\times10^{-7}$ | $1.19\times10^{-6}$ | $4.39\times10^{-6}$ | $1.86\times10^{-5}$ | $9.33\times10^{-5}$ | $2.29\times10^{-4}$ |
| $k_{cTt2}$ | $1.05\times10^{-9}$ | $1.94\times10^{-9}$ | $4.60\times10^{-9}$ | $1.19\times10^{-8}$ | $3.40\times10^{-8}$ | $1.09\times10^{-7}$ | $4.02\times10^{-7}$ | $8.27\times10^{-7}$ |
| $k_{tTt1}$ | $1.99\times10^{-6}$ | $4.84\times10^{-6}$ | $1.69\times10^{-5}$ | $6.70\times10^{-5}$ | $3.07\times10^{-4}$ | $1.66\times10^{-3}$ | $1.10\times10^{-2}$ | $3.12\times10^{-2}$ |
| $k_{tTt2}$ | $5.81\times10^{-8}$ | $1.24\times10^{-7}$ | $3.60\times10^{-7}$ | $1.16\times10^{-6}$ | $4.26\times10^{-6}$ | $1.80\times10^{-5}$ | $8.96\times10^{-5}$ | $2.19\times10^{-4}$ |
| $k_{tCt1}$ | $3.10\times10^{-6}$ | $7.83\times10^{-6}$ | $2.88\times10^{-5}$ | $1.21\times10^{-4}$ | $5.88\times10^{-4}$ | $3.40\times10^{-3}$ | $2.42\times10^{-2}$ | $7.18\times10^{-2}$ |
| $k_{tCt1}$ | $7.77\times10^{-8}$ | $1.70\times10^{-7}$ | $5.13\times10^{-7}$ | $1.73\times10^{-6}$ | $6.58\times10^{-6}$ | $2.91\times10^{-5}$ | $1.53\times10^{-4}$ | $3.85\times10^{-4}$ |
| $k_{cCt1}$ | $1.02\times10^{-5}$ | $2.65\times10^{-5}$ | $1.02\times10^{-4}$ | $4.49\times10^{-4}$ | $2.32\times10^{-3}$ | $1.43\times10^{-2}$ | $1.09\times10^{-1}$ | $3.35\times10^{-1}$ |
| $k_{cCt1}$ | $5.48\times10^{-8}$ | $1.16\times10^{-7}$ | $3.37\times10^{-7}$ | $1.08\times10^{-6}$ | $3.94\times10^{-6}$ | $1.65\times10^{-5}$ | $8.21\times10^{-5}$ | $1.99\times10^{-4}$ |





**Table 2.** Relative rate constants of oxalic acid catalyzed $SO_3$ hydration with respect to water catalyzed $SO_3$ hydration at different altitudes.

| altitude (km) | 0 | 0 | 2 | 4 | 6 | 8 | 10 | 12 |
|---|---|---|---|---|---|---|---|---|
| P (bar) | 1.01325 | 1.01325 | 0.795 | 0.617 | 0.472 | 0.357 | 0.265 | 0.194 |
| T (K) | 298.15 | 288.15 | 275.15 | 262.17 | 249.19 | 236.22 | 223.25 | 216.65 |
| $k_{cTt1}/k_{w1}$ | $3.88\times10^4$ | $6.27\times10^4$ | $1.23\times10^5$ | $2.58\times10^5$ | $5.90\times10^5$ | $1.47\times10^6$ | $4.12\times10^6$ | $7.30\times10^6$ |
| $k_{cTt2}/k_{w2}$ | $3.36\times10^2$ | $3.83\times10^2$ | $4.63\times10^2$ | $5.70\times10^2$ | $7.22\times10^2$ | $9.41\times10^2$ | $1.28\times10^3$ | $1.52\times10^3$ |
| $k_{tTt1}/k_{w1}$ | $1.31\times10^6$ | $2.41\times10^6$ | $5.65\times10^6$ | $1.45\times10^7$ | $4.13\times10^7$ | $1.31\times10^8$ | $4.84\times10^8$ | $9.95\times10^8$ |
| $k_{tTt2}/k_{w2}$ | $1.86\times10^4$ | $2.45\times10^4$ | $3.62\times10^4$ | $5.58\times10^4$ | $9.04\times10^4$ | $1.55\times10^5$ | $2.85\times10^5$ | $4.00\times10^5$ |
| $k_{tCt1}/k_{w1}$ | $2.05\times10^6$ | $3.90\times10^6$ | $9.62\times10^6$ | $2.61\times10^7$ | $7.91\times10^7$ | $2.69\times10^8$ | $1.07\times10^9$ | $2.29\times10^9$ |
| $k_{tCt2}/k_{w2}$ | $2.49\times10^4$ | $3.37\times10^4$ | $5.16\times10^4$ | $8.26\times10^4$ | $1.40\times10^5$ | $2.51\times10^5$ | $4.88\times10^5$ | $7.06\times10^5$ |
| $k_{cCt1}/k_{w1}$ | $6.73\times10^6$ | $1.32\times10^7$ | $3.42\times10^7$ | $9.72\times10^7$ | $3.12\times10^8$ | $1.13\times10^9$ | $4.80\times10^9$ | $1.07\times10^{10}$ |
| $k_{cCt2}/k_{w2}$ | $1.76\times10^4$ | $2.30\times10^4$ | $3.39\times10^4$ | $5.19\times10^4$ | $8.37\times10^4$ | $1.43\times10^5$ | $2.61\times10^5$ | $3.65\times10^5$ |





**Table 3.** Conformational population for oxalic acid conformers at different altitudes.

| altitude (km) | 0 | 0 | 2 | 4 | 6 | 8 | 10 | 12 |
|---|---|---|---|---|---|---|---|---|
| P (bar) | 1.01325 | 1.01325 | 0.795 | 0.617 | 0.472 | 0.357 | 0.265 | 0.194 |
| T (K) | 298.15 | 288.15 | 275.15 | 262.17 | 249.19 | 236.22 | 223.25 | 216.65 |
| cTc | 96.51 % | 97.06 % | 97.69 % | 98.22 % | 98.66 % | 99.01 % | 99.30 % | 99.42 % |
| cTt | 2.27 % | 1.98 % | 1.62 % | 1.30 % | 1.02 % | 0.78 % | 0.57 % | 0.48 % |
| tTt | 0.71 % | 0.56 % | 0.41 % | 0.29 % | 0.20 % | 0.13 % | 0.08 % | 0.06 % |
| tCt | 0.48 % | 0.38 % | 0.26 % | 0.18 % | 0.11 % | 0.07 % | 0.04 % | 0.03 % |
| cCt | 0.03 % | 0.02 % | 0.02 % | 0.01 % | 0.01 % | 0.01 % | 0.01 % | 0.01 % |



**Table 4.** Concentrations (in molecule cm$^{-3}$) of oxalic acid conformers involving in SO$_3$ hydration and water at different altitudes.

| altitude (km) | 0 | 0 | 2 | 4 | 6 | 8 | 10 | 12 |
|---|---|---|---|---|---|---|---|---|
| P (bar) | 1.01325 | 1.01325 | 0.795 | 0.617 | 0.472 | 0.357 | 0.265 | 0.194 |
| T (K) | 298.15 | 288.15 | 275.15 | 262.17 | 249.19 | 236.22 | 223.25 | 216.65 |
| cTt | $4.59\times10^7$ | $4.13\times10^7$ | $2.78\times10^7$ | $1.82\times10^7$ | $1.15\times10^7$ | $6.98\times10^6$ | $4.05\times10^6$ | $2.58\times10^6$ |
| tTt | $1.42\times10^7$ | $1.18\times10^7$ | $7.08\times10^6$ | $4.08\times10^7$ | $2.25\times10^6$ | $1.17\times10^6$ | $5.74\times10^5$ | $3.33\times10^5$ |
| tCt | $9.72\times10^6$ | $7.84\times10^6$ | $4.53\times10^6$ | $2.50\times10^6$ | $1.31\times10^6$ | $6.51\times10^5$ | $3.01\times10^5$ | $1.69\times10^5$ |
| cCt | $6.35\times10^5$ | $4.82\times10^5$ | $2.56\times10^5$ | $1.29\times10^5$ | $6.13\times10^4$ | $2.72\times10^4$ | $1.11\times10^4$ | $5.80\times10^3$ |
| H$_2$O | [a]$5.18\times10^{17}$ | $1.92\times10^{17}$ | $9.57\times10^{16}$ | $3.47\times10^{16}$ | $1.22\times10^{16}$ | $3.80\times10^{15}$ | $5.97\times10^{14}$ | $1.18\times10^{14}$ |

[a]The water vapor concentration at 0 km and 298.15 K are obtained from (Torrent-Sucarrat et al., 2012). Other water vapor concentrations are taken from U.S. Standard Atmosphere, 1976.





**Table 5.** Relative rate of $SO_3$ hydration reaction catalyzed by oxalic acid and by water at different altitudes.

| altitude (km) | 0 | 0 | 2 | 4 | 6 | 8 | 10 | 12 |
|---|---|---|---|---|---|---|---|---|
| P (bar) | 1.01325 | 1.01325 | 0.795 | 0.617 | 0.472 | 0.357 | 0.265 | 0.194 |
| T (K) | 298.15 | 288.15 | 275.15 | 262.17 | 249.19 | 236.22 | 223.25 | 216.65 |
| $^a\nu_{SO3\cdots H2O+OA}/\nu_{SO3\cdots H2O+H2O}$ | $8.62\times10^{-5}$ | $3.54\times10^{-4}$ | $1.00\times10^{-3}$ | $4.08\times10^{-3}$ | $1.82\times10^{-2}$ | $9.74\times10^{-2}$ | 1.12 | 6.78 |

$^a$The rate $\nu_{SO3\cdots H2O+OA}$ represents the sum of reaction rate for $SO_3\cdots H_2O$ complex with four oxalic acid conformers (cTt, tTt, tCt, cCt). The rate ($\nu_{SO3\cdots H2O+cTt}/\nu_{SO3\cdots H2O+H2O}$, $\nu_{SO3\cdots H2O+tTt}/\nu_{SO3\cdots H2O+H2O}$, $\nu_{SO3\cdots H2O+tCt}/\nu_{SO3\cdots H2O+H2O}$, $\nu_{SO3\cdots H2O+cCt}/\nu_{SO3\cdots H2O+H2O}$) are shown in Table S10.