# Peer review of "Understanding the catalytic role of oxalic acid in the SO3 hydration to form H2SO4 in the atmosphere"

_Atmospheric Chemistry and Physics, 2018_

## Referee Comment (RC1) · Anonymous Referee #1 · 5 Sep 2018

Using quantum chemical calculations combined with high level ab initio method; the authors studied the catalytic ability of the most common dicarboxylic acid in the atmosphere - oxalic acid - for the hydration reaction of SO3. Further, taking the real atmosphere into consideration, they found that oxalic acid-catalyzed hydration reaction can compete with the water-catalyzed reaction in the upper troposphere, which has certain significance for the formation of H2SO4 in the atmosphere. The work is performed with care and I believe it can be published after the following concerns are fully addressed. Major comments: 1. In previous paper (Hazra et al, J. Am. Chem. Soc. 2011, 133, 17444), hydrolysis of SO3 catalyzed by formic acid in the gas phase has been studied and the result shows a near barrierless mechanism for sulfuric acid

formation. Moreover, we note that formic acid is considered to be the most abundant carboxylic acid, ubiquitous in the atmosphere (Millet et al. Atmos. Chem. Phys., 2015, 15, 6283; Bannan et al. J. Geophys. Res. - Atmos, 2017, 122, 488). Thus, to make this story more interesting, I think the authors should highlight the specific characteristic of oxalic acid compared with formic acid, and add more discussion about the advantages of oxalic acid acting as a catalyst. 2. For method and basis set of the single point energy, the CCSD(T) method with the 6-311++G(3df,3pd) is not a good match, I think it is available using at least cc-pVTZ level. If the author prefer the 6-311++G(3df,3pd) basis set when using CCSD(T) to perform the single point energy, please give the corresponding reasons. 4.In Section 3.1: it is better to compare the results with previous studies to enrich the text. Minor comments: 1. Page1, line 12 (Abstract), "can involve in" should be "can be involved in". 2. Page1, line 16 (Abstract), "the rate of SO3 hydration" should be "the rates of SO3 hydration". 3. Page 2, line 5: is it better to change the second "can" to "thus"? 4. Page 2, line 11: "firstly is" should be "is firstly". 5. Page 2, line 11: "rearrange" should be "rearranges". 6. Page 2, line 14: "takes" should be "take" 7. Page 2, line 17: "reduce" should be "reduces" 8. Page 3, line 15: "for" should be "of" 9. Page 3, line 17: "... the hydration of SO3 in the second water ..." should be "... the hydration of SO3 with the second water ...". 10. Page 3, line 20: "reactant, complex, transition state" should be "reactants, complexes, transition states" 11. Page 4, lines 9-10: "..., the electronic energies based on the ..., while the partition functions obtained from ..." should be "..., the electronic energies were based on the ..., while the partition functions were obtained from ..." 12. Page 4, line 14: "..., it can conclude that the ..." should be "..., it can be concluded that the ..." 13. Page 5: "The corresponding rate constants are that" should be "The corresponding reaction rates are that". 14. Page 6, line 10: "react" should be "reacts" 15. Page 6, line 11: "rearrange" should be "rearranges" and "begin" should be "begins" 16. Page 7, line 10: "Table S2-S5" should be "Tables S2-S5" 17. Page 10, line 23: is it better to change "real" to "potential" 18. Page 11, line 3: is it better to change "The energy barrier of hydration reaction of SO3 is about or below 1 kcal mol-1" to "Other conformers can

catalyze the hydration reaction and the corresponding energy barrier is a little higher or less than 1 kcal mol-1" 19. Page 11, line 4: add "that" after "signify" 20. Page 11, line 5: add "from the view of barrier" after "H2SO4" 21. Page 22, the caption of Figure 4. Please add the unit of hydrogen bond length.

Please also note the supplement to this comment:
https://www.atmos-chem-phys-discuss.net/acp-2018-490/acp-2018-490-RC1-supplement.pdf

———————————————————

---

## Referee Comment (RC2) · Anonymous Referee #3 · 14 Nov 2018

This paper presents quantum chemical calculations for the catalytic role of oxalic acid in the SO3 hydration to form H2SO4. This topic is appropriate for the ACP journal, and the results of this study are somewhat interesting. However, this work is incomplete and the results are poorly presented. Thus, significant revision is needed before this paper can be considered for publication in ACP.

From my perspective, the authors should strive to address the following points:

1. The authors tried to state "the oxalic acid-catalyzed SO3 hydration can compete with water-catalyzed SO3 hydration". To be frank, I cannot assess whether this is correct from the current results. The presence of oxalic acid exactly enhances the rate

constants for the hydration of SO3, but I know that the concentration of water is much great larger than that of oxalic acid in the atmosphere. So I have reason to believe that the half-life of water-catalyzed SO3 hydration is much great smaller than that of the oxalic acid-catalyzed SO3 hydration. If so, maybe "the oxalic acid-catalyzed SO3 hydration can compete with water-catalyzed SO3 hydration" is not correct, except that the authors can prove it.

2. I remain unconvinced about whether the authors clearly know what the addition, decomposition, isomerization, and abstraction reactions are. p2: "Some addition, decomposition, isomerization, and abstraction reactions also are the important HAT reaction in the atmosphere." These reactions are parallel, rather than a containment relationship!

3. The authors gave the binding energies of water dimers, RC1, PC1, etc, but didn't discuss them. I think they are not any function in this paper? So either the author add some discussion about the binding energies or delete the binding energies. Furthermore, the authors also define what is the binding energies.

4. In the paper, the authors named the title of each section as "Water-catalyzed hydration reaction of SO3", "Oxalic acid-catalyzed hydration reaction of SO3" and so on. However, I cannot find any results about "catalyzed" in these sections.

5. The authors pointed out that oxalic acid as one of the hydrogen donors and/or acceptors could catalyze the hydration reaction of SO3, through the formation of two-point hydrogen bond. Is it possible that all the hydration reaction of SO3 could be enhanced in the presence of any species with the formation the two-point hydrogen bond? So, the authors should summary the relationship with these species and the enhancement of the hydration reaction of SO3.

6. I also cannot assess the key ingredient: whether the details of the kinetic calculations are correct. As we know, limit value of atmospheric gas phase rate constant should not be not more than 10-9 cm3 molecule-1 s-1, especially for the reaction with barrier.

However, the calculated rate constants in this work are only 10-5, 10-6, and 10-10 cm3 molecule-1 s-1 for some reactions. So, I think the authors should check their methods and results.

7. The authors mentioned that "hydroperoxy radical, formic acid, sulfuric acid (Torrent-Sucarrat et al., 2012), nitric acid and ammonia have been reported to replace the second water to catalyze the hydration reaction of SO3 ." I think the authors should compare the catalytic effect of these species with oxalic acid?

8. The paper is required to be revised by native English speakers.

---

## Author Comment (AC1) · 14 Dec 2018

We thank the reviewer for your helpful comments. We largely agree with your constructive advice, and have carefully revised our manuscript. In the following, the detailed response to your comments is listed. All changes in revised manuscript have been marked with green color. In the responses below, responses are marked in purple color. And the corresponding content in revised manuscript still is shown in green in these responses.

**Author Responses to Anonymous Referee #1 (RC1)**

Using quantum chemical calculations combined with high level ab initio method; the authors studied the catalytic ability of the most common dicarboxylic acid in the atmosphere - oxalic acid - for the hydration reaction of SO3. Further, taking the real atmosphere into consideration, they found that oxalic acid-catalyzed hydration reaction can compete with the water-catalyzed reaction in the upper troposphere, which has certain significance for the formation of $H_2SO_4$ in the atmosphere. The work is performed with care and I believe it can be published after the following concerns are fully addressed.

**Reply:** We appreciate the reviewer for your comments and advice. We have revised the manuscript on the basis of your comments and advice. The detail replies are described below.

Major comments:

1. In previous paper (Hazra et al, J. Am. Chem. Soc. 2011, 133, 17444), hydrolysis of SO3 catalyzed by formic acid in the gas phase has been studied and the result shows a near barrierless mechanism for sulfuric acid formation. Moreover, we note that formic acid is considered to be the most abundant carboxylic acid, ubiquitous in the atmosphere (Millet et al. Atmos. Chem. Phys., 2015, 15, 6283; Bannan et al. J. Geophys. Res. - Atmos, 2017, 122, 488). Thus, to make this story more interesting, I think the authors should highlight the specific characteristic of oxalic acid compared with formic acid, and add more discussion about the advantages of oxalic acid acting as a catalyst.

**Reply:** Thank you for your advice. Oxalic acid, as the prevalent dicarboxylic acid, really has some specific characteristic compared with formic acid. The studies have shown that dicarboxylic acid can enhance nucleation in two directions compared with momocarboxylic acid. Thus, in the oxalic

acid-catalyzed $SO_3$ hydration reaction, the post-reactive complex (PC) still has a free carboxylic group. The PC has the potential to continue to catalyze $SO_3$ hydration by the free carboxylic group. In addition, the result of the manuscript shows that the PC is stable with respect to the isolate oxalic acid conformers and $H_2SO_4$. Thus, the PC also has the potential to promote nucleation.

We also added more discussion about the advantages of oxalic acid acting as a catalyst. In this revised manuscript, we compared the catalytic effect of oxalic acid with the nitric acid, sulfuric acid, formic acid, ammonia.

The added contents are shown in revised manuscript with the marked green color. We also put these contents below:

In Introduction section, page 3, line 9 - 11: Opposite to monocarboxylic acids, dicarboxylic acids such as oxalic acid has been proved to enhance nucleation in two directions because of its two acid moieties (Xu and Zhang, 2012).

Page 9, the last paragraph: In addition, compared to formic acid (Millet et al., 2015;Bannan et al., 2017), the $SO_3$ hydration reaction catalyzed by oxalic acid display some specific characteristics. Both acids can obviously decrease the energy barrier of the hydration reaction. But because oxalic acid is a dicarboxylic acid, only one in the carboxylic groups participates in the hydration reaction, and the other is free. This characteristic indicates that post-reactive complexes ($PC_{tTt}$ and $PC_{tCt}$) can serve as a catalyst to continue to promote the $SO_3$ hydration. For these post-reactive complexes ($PC_{cTt}$, $PC_{tTt}$, $PC_{tCt}$ and $PC_{cCt}$), the above result has shown that these complexes are stable compared to the isolate $H_2SO_4$ and oxalic acid conformers. The free carboxylic group in these complexes can also provide the interaction site when these complexes interact with other species. The free carboxylic group of these stable post-reactive complexes indicates that these complexes have the potential to participate in nucleation.

Page 10 - 11, line 21 - 23 (page 10) and line 1 - 7 (page 11): As seen from Table2, at 298.15 K, the oxalic acid-catalyzed $SO_3$ hydration reaction is $10^3$ - $10^5$ faster than the corresponding water-catalyzed reaction ($k_{cTt1}/k_{w1}$: $1.53 \times 10^3$; $k_{tTt1}/k_{w1}$: $5.50 \times 10^4$; $k_{tCt1}/k_{w1}$: $9.70 \times 10^4$; $k_{cCt1}/k_{w1}$: $3.31 \times 10^5$). In order to evaluate the catalytic effect of oxalic acid, we also list the rate constant ratio between other species and water catalyzed $SO_3$ hydration reaction. At 298 K, the nitric acid catalyzed rate constant is just 1.19 times larger than water catalyzed rate constant (Long et al., 2013). When sulfuric acid functions as an autocatalyst, the value of rate constant ratio is around $10^2$ (Torrent-Sucarrat et al., 2012). When the formic acid acts as a catalyst, its rate constant is $10^4$ greater than that of water-catalyzed $SO_3$

hydration (Hazra and Sinha, 2011). The rate constant for ammonia catalyzed $SO_3$ hydration is 7 orders of magnitude larger than that for water catalyzed rate constant at 298 K (Bandyopadhyay et al., 2017). These results indicate that the catalytic effect of oxalic acid in $SO_3$ hydration reaction is better than nitric acid and sulfuric acid, and similar to formic acid, but weaker than ammonia.

2. For method and basis set of the single point energy, the CCSD(T) method with the 6-311++G(3df,3pd) is not a good match, I think it is available using at least cc-pVTZ level. If the author prefer the 6-311++G(3df,3pd) basis set when using CCSD(T) to perform the single point energy, please give the corresponding reasons.

**Reply:** Thank you for your comment and advice. We have recalculated the single point energy at CCSD(T)/cc-pV(T+d)Z level. As the influence of single point energy is overall, the potential energy profile and the rate constant are changed after using the new theoretic level (CCSD(T)/cc-pV(T+d)Z). For potential energy, we do not mark the changed data in the manuscript because all data are changed. Although the changed data are not marked in the revised manuscript, old and new figures are put below for showing the change:

The old Figure 1:

[Figure]

**Figure 1.** Calculated potential energy profile for the hydration of $SO_3$ with the second water as a catalyst at the CCSD(T)/6-311++G(3df,3pd)//M06-2X/6-311++G(3df,3pd) level.

The new Figure 1:

[Figure]

**Figure 1.** Calculated potential energy profile for the hydration of $SO_3$ with the second water as a catalyst at the CCSD(T)/cc-pV(T+d)Z//M06-2X/6-311++G(3df,3pd) level.

The old Figure 3:

[Figure]

**When X = cTt,**
  a = -6.48    b = -9.05    c = -20.29    d = -18.92    e = -33.40    f = -14.86 (X'=cCt)

**When X = tTt,**
  a = -6.48    b = -8.76    c = -21.58    d = -21.14    e = -35.33    f = -17.27 (X'=tCt)

**When X = tCt,**
  a = -6.48    b = -8.93    c = -22.04    d = -21.76    e = -36.05    f = -18.32 (X'=tTt)

**When X = cCt,**
  a = -6.48    b = -9.92    c = -22.53    d = -22.27    e = -37.60    f = -20.73 (X'=cTt)

**Figure 3.** Calculated potential energy profile for the hydration of $SO_3$ with oxalic acid conformers (cTt, tTt, tCt and cCt) as catalysts at the CCSD(T)/6-311++G(3df,3pd) //M06-2X/6-311++G(3df,3pd) level.

The new Figure 3:

[Figure]

When X = cTt,
a = -7.38   b = -10.25   c = -21.20   d = -20.01   e = -34.52   f = -16.74 (X'=cCt)

When X = tTt,
a = -7.38   b = -9.92   c = -22.51   d = -22.27   e = -36.56   f = -19.22 (X'=tCt)

When X = tCt,
a = -7.38   b = -10.09   c = -23.01   d = -22.96   e = -37.45   f = -20.47 (X'=tTt)

When X = cCt,
a = -7.38   b = -11.10   c = -23.48   d = -23.49   e = -39.10   f = -22.95 (X'=cTt)

**Figure 3.** Calculated potential energy profile for the hydration of $SO_3$ with oxalic acid conformers (cTt, tTt, tCt and cCt) as catalysts at the CCSD(T)/cc-pV(T+d)Z //M06-2X/6-311++G(3df,3pd) level.

For the kinetics analysis, to make the rate constant more meaningful, we adjusted the entrance for rate constant calculation from $SO_3$ + $H_2O\cdots H_2O$ (or $SO_3\cdots H_2O$ + $H_2O$) and $SO_3$ + $H_2O\cdots OA$ (or $SO_3\cdots OA$ + $H_2O$) to $SO_3$ + $H_2O$ + $H_2O$ and $SO_3$ + $H_2O$ + OA. And the adjustment will not affect the result of the rate comparison. The adjustment caused the great changes for Table 1, which lists the rate constant. Using the new level of theory to calculate the single point energy, it can be found from the old and new Table 5 that the relative rate decreases from 6.78 to around 0.1 at 12 km altitude. It still indicates that the oxalic acid can play a significant role in enhancing $SO_3$ hydration to $H_2SO_4$. The correspond content in revised manuscript also are described as follows:

Page 12, line 1 - 5: When the altitude increases to 10 km, the oxalic acid-catalyzed reaction is just 2 orders of magnitude slower than water-catalyzed reaction. At 12 km altitude, the water-catalyzed hydration reaction is approximately 10 times faster than the oxalic acid-catalyzed $SO_3$ hydration. To sum up, the comparison of relative rate shows that, in the upper troposphere, the oxalic acid can play a

significant role in enhancing $SO_3$ hydration to $H_2SO_4$.

We also put the changed tables as follows:

The old Table 1:

[revised manuscript text omitted]

[a]The rate $v_{SO3\cdots H2O+OA}$ represents the sum of reaction rate for $SO_3\cdots H_2O$ complex with four oxalic acid conformers (cTt, tTt, tCt, cCt). The rate ($v_{SO3\cdots H2O+cTt}/v_{SO3\cdots H2O+H2O}$, $v_{SO3\cdots H2O+tTt}/v_{SO3\cdots H2O+H2O}$, $v_{SO3\cdots H2O+tCt}/v_{SO3\cdots H2O+H2O}$, $v_{SO3\cdots H2O+cCt}/v_{SO3\cdots H2O+H2O}$) are shown in Table S10.

The new Table 5:

**Table 5.** Relative rate of $SO_3$ hydration reaction catalyzed by oxalic acid and by water at different altitudes.

| altitude (km) | 0 | 0 | 2 | 4 | 6 | 8 | 10 | 12 |
|---|---|---|---|---|---|---|---|---|
| P (bar) | 1.01325 | 1.01325 | 0.795 | 0.617 | 0.472 | 0.357 | 0.265 | 0.194 |
| T (K) | 298.15 | 288.15 | 275.15 | 262.17 | 249.19 | 236.22 | 223.25 | 216.65 |
| [a]$v_{OA1}/v_{w1}$ | $3.87\times10^{-6}$ | $1.43\times10^{-5}$ | $3.47\times10^{-5}$ | $1.20\times10^{-4}$ | $4.43\times10^{-4}$ | $1.95\times10^{-3}$ | $1.80\times10^{-2}$ | $9.53\times10^{-2}$ |

[a]The rate $v_{OA1}$ represents the sum of reaction rate for Reaction X1 (X=cTt, tTt, tCt, cCt). The rate ($v_{cTt1}/v_{w1}$, $v_{tTt1}/v_{w1}$, $v_{tCt1}/v_{w1}$, $v_{cCt1}/v_{w1}$) are shown in Table S10.

3. In Section 3.1: it is better to compare the results with previous studies to enrich the text.

**Reply:** Thank you for your advice. We have added the discussion about the comparison of the results with literatures. The added contents are shown in the revised manuscript (Page 7, line 8 - 14), which also are put below:

Page 7, line 9 - 13: Moreover, the binding energy of water dimer is consistent with its experimental value of $3.15 \pm 0.03$ kcal $mol^{-1}$ (Rocher-Casterline et al., 2011), and with theoretical results of 2.90 (Torrent-Sucarrat et al., 2012), 2.97 (Long et al., 2013), 3.14 (Hazra and Sinha, 2011) and 3.30 kcal $mol^{-1}$ (Klopper et al., 2000). In $SO_3 \cdots H_2O$ complex, the binding energy of 7.38 kcal $mol^{-1}$ in our paper agrees with that from theoretical estimates in the literature: 7.60 (Torrent-Sucarrat et al., 2012), 7.42 (Long et al., 2013), 7.25 (Hazra and Sinha, 2011) and 7.77 kcal $mol^{-1}$ (Long et al., 2012).

Page 7, line 15 - 16: The RC1 has the binding energy of 14.12 kcal $mol^{-1}$ relative to $SO_3+H_2O \cdots H_2O$, which is in accord with 13.60 (Torrent-Sucarrat et al., 2012) and 13.76 kcal $mol^{-1}$ (Long et al., 2013).

Minor comments:
1. Page1, line 12(Abstract), "can involve in" should be "can be involved in".

**Reply:** Thank you for pointing out our mistake. We have corrected this error. The corrected content can be seen from Abstract (Page 1, line 13), which also is shown as follows:

Some atmospheric species can be involved in and facilitate the reaction.

2. Page1, line 16 (Abstract), "the rate of SO3 hydration" should be "the rates of SO3 hydration".

**Reply:** We have corrected this error in the revised manuscript (Page 1, line 16). The new content is listed below:

Compared with the rates of $SO_3$ hydration reaction catalyzed by oxalic acid and water, it can be

found that, in the upper troposphere, the oxalic acid-catalyzed $SO_3$ hydration can play an important role in promoting the $SO_3$ hydration.

3. Page 2, line 5: is it better to change the second "can" to "thus"?

**Reply:** We have corrected this error in the revised manuscript (Page 2, line 5). The new content is listed below:

These atmospheric HAT reactions display a main feature that two-point hydrogen bond can occur and thus facilitate hydrogen atom transfer

4. Page 2, line 11: "firstly is" should be "is firstly".

**Reply:** We have corrected this error in the revised manuscript (Page 2, line 11). The new content is listed below:

For the reaction $SO_3+H_2O{\rightarrow}H_2SO_4$, the pre-reactive $SO_3{\cdots}H_2O$ complex is firstly formed, and the complex is then rearranged to produce $H_2SO_4$, which was proposed by Castleman et al (Holland and Castleman, 1978;Hofmann-Sievert and Castleman, 1984).

5. Page 2, line 11: "rearrange" should be "rearranges".

**Reply:** We have corrected this error in the revised manuscript (Page 2, line 11). The new content is listed below:

For the reaction $SO_3+H_2O{\rightarrow}H_2SO_4$, the pre-reactive $SO_3{\cdots}H_2O$ complex is firstly formed, and the complex is then rearranged to produce $H_2SO_4$, which was proposed by Castleman et al (Holland and Castleman, 1978;Hofmann-Sievert and Castleman, 1984).

6. Page 2, line 14:"takes" should be "take".

**Reply:** We have corrected this error in the revised manuscript (Page 2, line 13). The new content is listed below:

But the subsequent research found that this hydration reaction involving a single water molecule cannot take place in the atmosphere due to high energy barrier.

7. Page 2, line 17: "reduce" should be "reduces"

**Reply:** We have corrected this error in the revised manuscript (Page 2, line 17). The new content is listed below:

The promoting effect can be mainly attributed to the formation of the two-point hydrogen bond, which reduces the ring strain...

8. Page 3, line 15: "for" should be "of".

**Reply:** We have corrected this error in the revised manuscript (Page 3, line 17). The new content is listed below:

The rate constants of oxalic acid-catalyzed $SO_3$ hydration were calculated using the kinetics analysis, and compared with that of water-catalyzed hydration reaction.

9. Page 3, line 17: "... the hydration of SO3 in the second water ..." should be "... the hydration of SO3 with the second water ...".

**Reply:** We have corrected this error in the revised manuscript (Page 3, line 19). The new content is listed below:

...we evaluated the importance of the hydration process involving the oxalic acid relative to the hydration of $SO_3$ with the second water as a catalyst to form sulfuric acid

10. Page 3, line 20: "reactant, complex, transition state" should be "reactants, complexes, transition states".

**Reply:** We have corrected this error in the revised manuscript (Page 3, line 22). The new content is listed below:

The geometric structures, including all reactants, complexes, transition states and products, were

optimized using M06-2X method (Zhao and Truhlar, 2008) with 6-311++G(3df,3pd) basis set.

11. Page 4, lines 9-10: "..., the electronic energies based on the ..., while the partition functions obtained from ..." should be "..., the electronic energies were based on the ..., while the partition functions were obtained from ...".

**Reply:** We have corrected this error in the revised manuscript (Page 4, line 11 - 12). The new content is listed below:

In the kinetics analysis, the electronic energies were based on the CCSD(T)/cc-pV(T+d)Z level of theory, while the partition functions were obtained from the M06-2X/6-311++G(3df,3pd) level of theory.

12. Page 4, line 14: "..., it can conclude that the ..." should be "..., it can be concluded that the ...".

**Reply:** We have corrected this error in the revised manuscript (Page 4, line 16). The new content is listed below:

Base on the discussion in this paper, it can be concluded that the $SO_3$ hydration reactions begin with the formation of pre-reactive complex...

13. Page 5: "The corresponding rate constants are that" should be "The corresponding reaction rates are that".

**Reply:** We have corrected this error in the revised manuscript (Page 5, line 7). The new content is listed below:

The corresponding rates are that: ...

14. Page 6, line 10: "react" should be "reacts".

**Reply:** We have corrected this error in the revised manuscript (Page 7, line 2). The new content is listed below:

One is that water dimer reacts with $SO_3$ to obtain pre-reactive complex, then this complex is rearranged to form $H_2SO_4\cdots H_2O$ complex (channel 1); the other begins with the reaction of $SO_3\cdots H_2O$ complex with water, the following rearrangement is the same as the channel 1 (channel 2).

15. Page 6, line 11: "rearrange" should be "rearranges" and "begin" should be "begins".

**Reply:** We have corrected this error in the revised manuscript (Page 7, line 3). The new content is listed below:

One is that water dimer reacts with $SO_3$ to obtain pre-reactive complex, then this complex is rearranged to form $H_2SO_4\cdots H_2O$ complex (channel 1); the other begins with the reaction of $SO_3\cdots H_2O$ complex with water, the following rearrangement is the same as the channel 1 (channel 2).

16. Page 7, line 10: "Table S2-S5" should be "Tables S2-S5".

**Reply:** We have corrected this error in the revised manuscript (Page 8, line 4). The new content is listed below:

Energies, enthalpies and free energies of all relevant species for oxalic acid catalyzed hydration of $SO_3$ are summarized in Supplement (Tables S2 - S5).

17. Page 10, line 23: is it better to change "real" to "potential".

**Reply:** We have corrected this error in the revised manuscript (Page 12, line 15). The new content is listed below:

The main conclusion of this work is that oxalic acid, the most abundant dicarboxylic acid in the atmosphere, has the remarkable ability to catalyze $SO_3$ hydration to $H_2SO_4$, and has the potential impact on the $H_2SO_4$ formation in the atmosphere.

18. Page 11, line 3: is it better to change "The energy barrier of hydration reaction of SO3 is about or below 1 kcal mol-1" to "Other conformers can catalyze the hydration reaction and the corresponding energy barrier is a little higher or less than 1 kcal mol-1".

**Reply:** Thank you for your advice. We have changed this sentence in the revised manuscript (Page 12, line 20). We also put is below:

Other conformers can catalyze the hydration reaction and the corresponding energy barrier is a little higher or less than 1 kcal mol$^{-1}$.

19. Page 11, line 4: add "that" after "signify".

**Reply:** We have corrected this error in the revised manuscript and changed the word "signify" to "demonstrate" (Page 12, line 21). The new content is listed below:

The results demonstrate that oxalic acid has the higher catalytic ability than water for $SO_3$ hydration to form $H_2SO_4$.

20. Page 11, line 5: add "from the view of barrier" after "H2SO4".

**Reply:** Thank you for your advice. We combined the sentence with the previous sentence into a new sentence, and mentioned the two factors (rate constant and concentration). Because the rate constant contains the meaning from the view of barrier, we did not add the content. The new sentence is shown as follows:

Page 13, line 2 - 3: Based on the two factors, our calculation shows that, in the upper troposphere, the oxalic acid can play an important role in $SO_3$ hydration to form $H_2SO_4$.

21. Page 22, the caption of Figure 4. Please add the unit of hydrogen bond length.

**Reply:** Thank you for your advice. We have added the unit in the caption of Figure 4. The new content is listed below:

**Figure 4.** Optimized M06-2X/6-311++G(3df,3pd) structures of reactant complexes, pre-reactive complexes, transition states and post-reactive complexes for the oxalic acid-catalyzed $SO_3$ hydration reaction (distance unit: angstrom).

---

## Author Comment (AC2) · 14 Dec 2018

We thank the reviewer for your helpful comments. We largely agree with your constructive advice, and have carefully revised our manuscript. In the following, the detailed response to your comments is listed. All changes in revised manuscript have been marked with green color. In the responses below, responses are marked in purple color. And the corresponding content in revised manuscript still is shown in green in these responses.

**Author Responses to Anonymous Referee #3 (RC2)**

This paper presents quantum chemical calculations for the catalytic role of oxalic acid in the SO3 hydration to form H2SO4. This topic is appropriate for the ACP journal, and the results of this study are somewhat interesting. However, this work is incomplete and the results are poorly presented. Thus, significant revision is needed before this paper can be considered for publication in ACP.

**Reply:** We appreciate the reviewer for your comments and advice. We have revised the manuscript on the basis of your comments and advice. The detail replies are itemized below.

From my perspective, the authors should strive to address the following points:

1. The authors tried to state "the oxalic acid-catalyzed $SO_3$ hydration can compete with water-catalyzed $SO_3$ hydration". To be frank, I cannot assess whether this is correct from the current results. The presence of oxalic acid exactly enhances the rate constants for the hydration of $SO_3$, but I know that the concentration of water is much great larger than that of oxalic acid in the atmosphere. So I have reason to believe that the half-life of water-catalyzed $SO_3$ hydration is much great smaller than that of the oxalic acid-catalyzed $SO_3$ hydration. If so, maybe "the oxalic acid-catalyzed $SO_3$ hydration can compete with water-catalyzed $SO_3$ hydration" is not correct, except that the authors can prove it.

**Reply:** The expression "the oxalic acid-catalyzed $SO_3$ hydration can compete with water-catalyzed $SO_3$ hydration" is really inappropriate. We have changed the expression to "the oxalic acid-catalyzed $SO_3$ hydration can play an important role in promoting the $SO_3$ hydration". In the oxalic acid or water catalyzed $SO_3$ hydration reaction, two factors (rate constant and concentration) affect the reaction rate. As shown in Equation 11 and Equation 12 in the manuscript, the comparison of the rate for oxalic acid

and water catalyzed $SO_3$ hydration is equal to the product of rate constant ratio ($k_{OA}/k_{water}$) and concentration ratio ([OA]/[water]). Although the concentration of water is great larger than that of oxalic acid, on certain conditions, the higher catalytic effect of oxalic acid for $SO_3$ hydration (higher rate constant) can compensate for the highly concentration difference between water vapor and oxalic acid. According to the analysis about the two factors, it has shown that the oxalic acid-catalyzed reaction just is 2 orders of magnitude slower than water-catalyzed reaction at 10 km altitude, and at 12 km altitude, the water-catalyzed hydration reaction is approximately 10 times faster than the oxalic acid-catalyzed $SO_3$ hydration. These results indicate that the oxalic acid-catalyzed $SO_3$ hydration can play an important role in promoting the $SO_3$ hydration in the upper troposphere. However, the expression "the oxalic acid-catalyzed $SO_3$ hydration can compete with water-catalyzed $SO_3$ hydration", especially using the word "compete", is inappropriate. We have changed it in the revised manuscript, which also is put below:

Abstract, Page 1, line 16 - 18: Compared with the rates of $SO_3$ hydration reaction catalyzed by oxalic acid and water, it can be found that, in the upper troposphere, the oxalic acid-catalyzed $SO_3$ hydration can play an important role in promoting the $SO_3$ hydration.

Conclusion, Page 13, line 2 - 3: Based on the two factors, our calculation shows that, in the upper troposphere, the oxalic acid can play an important role in $SO_3$ hydration to form $H_2SO_4$.

2. I remain unconvinced about whether the authors clearly know what the addition, decomposition, isomerization, and abstraction reactions are. p2: "Some addition, decomposition, isomerization, and abstraction reactions also are the important HAT reaction in the atmosphere." These reactions are parallel, rather than a containment relationship!

**Reply:** Thank you for your comments. Using the sentence, we want to express that the HAT process in the atmosphere can occur in some addition, decomposition, isomerization, and abstraction reactions. We know that these reactions are parallel. We express our thought unclearly, and we thank you for pointing out this. We have corrected the sentence in the revised manuscript. The new sentence is also shown below:

 The hydrogen atom transfer process can also be found in some addition (Steudel, 1995;Williams et al., 1983;Courmier et al., 2005;Zhang and Zhang, 2002), decomposition (Rayez et al., 2002;Kumar and Francisco, 2015;Gutbrod et al., 1996), isomerization (Zheng and Truhlar, 2010;Atkinson, 2007), and abstraction reactions (Ji et al., 2013;Ji et al., 2017).

3. The authors gave the binding energies of water dimers, RC1, PC1, etc, but didn't discuss them. I think they are not any function in this paper? So either the author add some discussion about the binding energies or delete the binding energies. Furthermore, the authors also define what is the binding energies.

**Reply:** Thank you for your comments. We have added some contents to compare binding energies of water dimers, $SO_3 \cdots H_2O$ complex and RC1 in our manuscript with that in the previous paper, and have deleted the content about the binding energies of PC1. For the binding energies, it always is used in the quantum chemical calculations to describe the reaction pathway.

The added contents are as follows:

Page 7, line 9 - 13: Moreover, the binding energy of water dimer is consistent with its experimental value of $3.15 \pm 0.03$ kcal mol$^{-1}$ (Rocher-Casterline et al., 2011), and with theoretical results of 2.90 (Torrent-Sucarrat et al., 2012), 2.97 (Long et al., 2013), 3.14 (Hazra and Sinha, 2011) and 3.30 kcal mol$^{-1}$ (Klopper et al., 2000). In $SO_3 \cdots H_2O$ complex, the binding energy of 7.38 kcal mol$^{-1}$ in our paper agrees with that from theoretical estimates in the literature: 7.60 (Torrent-Sucarrat et al., 2012), 7.42 (Long et al., 2013), 7.25 (Hazra and Sinha, 2011) and 7.77 kcal mol$^{-1}$ (Long et al., 2012).

Page 7, line 15 - 16: The RC1 has the binding energy of 14.12 kcal mol$^{-1}$ relative to $SO_3+H_2O \cdots H_2O$, which is in accord with 13.60 (Torrent-Sucarrat et al., 2012) and 13.76 kcal mol$^{-1}$ (Long et al., 2013).

For PC1, we have deleted the content about binding energies (old page 6, line 20, and old page 7 1 - 2). The deleted contents are list with strikethrough as follows:

4. In the paper, the authors named the title of each section as "Water-catalyzed hydration reaction of $SO_3$", "Oxalic acid-catalyzed hydration reaction of $SO_3$" and so on. However, I cannot find any results about "catalyzed" in these sections.

**Reply:** We have added the description about the catalyzed process. For water-catalyzed hydration reaction of $SO_3$, the added content can be found in Page 7, line 4 - 6, which also is shown as follows:

In the hydration process, the additional water molecule serves as a catalyst that can promote the reaction by making a bridge in the hydrogen atom transfer from water to $SO_3$.

For oxalic acid-catalyzed hydration reaction of $SO_3$, we have written the new content to describe the catalyzed process (Page 8, line 10 - 11), which also is put below:

In this process, oxalic acid serves as a catalyst to promote $SO_3$ hydration reaction by making a bridge when the hydrogen atom transfers from $H_2O$ to $SO_3$.

5. The authors pointed out that oxalic acid as one of the hydrogen donors and/or acceptors could catalyze the hydration reaction of $SO_3$, through the formation of twopoint hydrogen bond. Is it possible that all the hydration reaction of $SO_3$ could be enhanced in the presence of any species with the formation the two-point hydrogen bond? So, the authors should summary the relationship with these species and the enhancement of the hydration reaction of $SO_3$.

**Reply:** Thank you for your comments and advice. All species, which can form the two-point hydrogen bond with $SO_3$ and $H_2O$, have the potential to catalyze $SO_3$ hydration reaction. However, the enhancement effect of these species on $SO_3$ hydration reaction in the atmosphere needs to evaluate. Although some species can catalyze $SO_3$ hydration, the catalytic effect of the species is not enough to compensate for the very high difference in the concentration of water vapor and the species. For these species, it is of minor importance in the atmosphere for enhancing $SO_3$ hydration.

We accept your advice. We summarized the relationship with these species and the enhancement of the hydration reaction of $SO_3$ in the revised manuscript. The corresponding contents also are shown as follows:

Page 12, line 6 - 12: It has been shown that some species including nitric acid, sulfuric acid, formic acid, ammonia, hydroperoxy radical and oxalic acid in our study can catalyze the $SO_3$ hydration reaction. By forming two-point hydrogen bond, these species can make a bridge to promote the hydration reaction. It may be concluded that, as long as the species can form two-point hydrogen bond with water molecule and $SO_3$, it has the potential to promote the $SO_3$ hydration reaction. However, the real atmospheric importance about the species-catalyzed $SO_3$ hydration reaction needs to be evaluated. That is, compared to water catalyzed $SO_3$ hydration reaction, the species must have the sufficient catalytic effect, leading to the increase of rate constants, so as to compensate for the highly concentration difference between water vapor and the species.

6. I also cannot assess the key ingredient: whether the details of the kinetic calculations are correct. As we know, limit value of atmospheric gas phase rate constant should not be not more than $10^{-9}$ cm3 molecule$^{-1}$ s$^{-1}$, especially for the reaction with barrier. However, the calculated rate constants in this work are only $10^{-5}$, $10^{-6}$, and $10^{-10}$ cm$^3$ molecule$^{-1}$ s$^{-1}$ for some reactions. So, I think the authors should check their methods and results.

**Reply:** For the rate constant, we will firstly explain the reason for the result and then describe how to adjust it. When a catalyst occurs, the process of $SO_3$ hydration reaction can be formulated as (M represents $H_2O$ or oxalic acid conformers in this manuscript):

$$channel\,1:$$

$$SO_3 + H_2O \rightleftharpoons SO_3 \cdots H_2O \qquad (1a)$$

$$SO_3 \cdots H_2O + M \rightleftharpoons SO_3 \cdots M \cdots H_2O \rightarrow H_2SO_4 \cdots M \quad (1b)$$

$$channel\,2:$$

$$M + H_2O \rightleftharpoons M \cdots H_2O \qquad (2a)$$

$$SO_3 + M \cdots H_2O \rightleftharpoons SO_3 \cdots M \cdots H_2O \rightarrow H_2SO_4 \cdots M \quad (2b)$$

In the initial manuscript, rate calculation started from (1b) and (2b), and (1a) and (1b) are not considered. The reason is that the rate from (1b) can be described as $v = k[SO_3 \cdots H_2O][M]$. And the relative rate between oxalic acid and water catalyzed $SO_3$ hydration is $v_{OA}/v_{water} = k_{OA}[SO_3 \cdots H_2O][OA]/k_{water}[SO_3 \cdots H_2O][H2O] = k_{OA}[OA]/k_{water}[H2O]$. The concentration of $SO_3 \cdots H_2O$ complex can be eliminated.

Using the method above, the calculated rate constant is not the rate constant of the whole hydration process. So, the rate constant is larger than the limit value. As we focus on the relative rate, when reaction processes (1b) and (2b) are selected as the entrance of rate calculation, we can also obtain the relative rate so as to evaluate the importance of oxalic acid. This is the reason that we select the (1b) and (2b) as the entrance of rate calculation, and that the rate constant is larger than the limit value.

But for making the rate constant more meaningful, we recalculated rate constants of the whole hydration process (for channel 1, including (1a) and (1b); for channel, including (2a) and (2b)). The method has been adjusted in the revised manuscript. The changed content also is shown below:

Page 4, line 18 - 19:

$$A + B \underset{k_{-1}}{\overset{k_1}{\rightleftarrows}} C$$
$$C + D \underset{k_{-2}}{\overset{k_2}{\rightleftarrows}} pre-reactive\ complex \xrightarrow{k_{uni}} post-reactive\ complex \qquad (1)$$

Page 5, line 1:

$$v = \frac{k_1}{k_{-1}} \frac{k_2}{k_{-2}} k_{uni} [A][B][D] = K_{eq1} K_{eq2} k_{uni} [A][B][D] \qquad (2)$$

Page 5, line 5 - 10:

*Recation* 1:

$$SO_3 + H_2O \underset{k_{-1}}{\overset{k_1}{\rightleftarrows}} SO_3 \cdots H_2O \qquad (3)$$

$$SO_3 \cdots H_2O + H_2O \underset{k_{-2}^{w1}}{\overset{k_2^{w1}}{\rightleftarrows}} SO_3 \cdots H_2O \cdots H_2O \xrightarrow{k_{uni\_w}} H_2SO_4 \cdots H_2O$$

*Reaction* 2:

$$H_2O + H_2O \underset{k_{-1}^{w2}}{\overset{k_1^{w2}}{\rightleftarrows}} H_2O \cdots H_2O \qquad (4)$$

$$SO_3 + H_2O \cdots H_2O \underset{k_{-2}^{w2}}{\overset{k_2^{w2}}{\rightleftarrows}} SO_3 \cdots H_2O \cdots H_2O \xrightarrow{k_{uni\_w}} H_2SO_4 \cdots H_2O$$

The corresponding rates are that:

$$v_{w1} = \frac{k_1}{k_{-1}} \frac{k_2^{w1}}{k_{-2}^{w1}} k_{uni\_w} [SO_3][H_2O][H_2O] = K_{eq1} K_{eq2}^{w1} k_{uni\_w} [SO_3][H_2O][H_2O] = k_{w1} [SO_3][H_2O][H_2O] \quad (5)$$

$$v_{w2} = \frac{k_1^{w2}}{k_{-1}^{w2}} \frac{k_2^{w2}}{k_{-2}^{w2}} k_{uni\_w} [SO_3][H_2O][H_2O] = K_{eq1}^{w2} K_{eq2}^{w2} k_{uni\_w} [SO_3][H_2O][H_2O] = k_{w2} [SO_3][H_2O][H_2O] \quad (6)$$

Page 5, line 12 - 14 and Page 6, line 1 - 2:

*Reaction X1:*

$$SO_3 + H_2O \underset{k_{-1}}{\overset{k_1}{\rightleftarrows}} SO_3 \cdots H_2O \tag{7}$$

$$SO_3 \cdots H_2O + X \underset{k_{-2}^{X1}}{\overset{k_2^{X1}}{\rightleftarrows}} SO_3 \cdots H_2O \cdots X \xrightarrow{k_{uni\_X}} H_2SO_4 \cdots X'$$

*Reaction X2:*

$$H_2O + X \underset{k_{-1}^{X2}}{\overset{k_1^{X2}}{\rightleftarrows}} H_2O \cdots X \tag{8}$$

$$SO_3 + H_2O \cdots X \underset{k_{-2}^{X2}}{\overset{k_2^{X2}}{\rightleftarrows}} SO_3 \cdots H_2O \cdots X \xrightarrow{k_{uni\_X}} H_2SO_4 \cdots X'$$

Page 6, line 5 - 13:

$$v_{X1} = \frac{k_1}{k_{-1}} \frac{k_2^{X1}}{k_{-2}^{X1}} k_{uni\_X}[SO_3][H_2O][X] = K_{eq1} K_{eq2}^{X1} k_{uni\_X}[SO_3][H_2O][X] = k_{X1}[SO_3][H_2O][X] \tag{9}$$

$$v_{X2} = \frac{k_1^{X2}}{k_{-1}^{X2}} \frac{k_2^{X2}}{k_{-2}^{X2}} k_{uni\_X}[SO_3][H_2O][X] = K_{eq1}^{X2} K_{eq2}^{X2} k_{uni\_X}[SO_3][H_2O][X] = k_{X2}[SO_3][H_2O][X] \tag{10}$$

To assess the importance of oxalic acid in $SO_3$ hydration to $H_2SO_4$ in the atmosphere, the relative rate can be used as:

$$\frac{v_{X1}}{v_{w1}} = \frac{k_{X1}[SO_3][H_2O][X]}{k_{w1}[SO_3][H_2O][H_2O]} = \frac{k_{X1}[X]}{k_{w1}[H_2O]} \tag{11}$$

$$\frac{v_{X2}}{v_{w2}} = \frac{k_{X2}[SO_3][H_2O][X]}{k_{w2}[SO_3][H_2O][H_2O]} = \frac{k_{X2}[X]}{k_{w2}[H_2O]} \tag{12}$$

It can easily be inferred from these equations that the rate constants are same for Reaction 1 ($k_{w1}$) and Reaction 2 ($k_{w2}$), as well as for Reaction X1 ($k_{X1}$) and Reaction X2 ($k_{X2}$) (for proof, see Supplement, Text S1). Thus, the relative rate values in Equation 11 and Equation 12 are same. Base on these reasons, we will only compare the relative rate $v_{X1}/v_{w1}$ in this paper.

In addition, we also recalculated the single point energy using the level of theory (CCSD(T)/cc-pV(T+d)Z) so as to obtain the more accurate result. By adjusting the method for the rate calculation and changing the energy calculation method, we can obtain more meaningful and accurate results. New tables about rate constants, relative rate constants and relative rate are shown in revised manuscript (Table 1, Table 2 and Table 5). We also put the old and new tables below:

The old Table 1:

**Table 1.** Rate constants (in cm$^3$ molecule$^{-1}$ s$^{-1}$) of $SO_3$ hydration reaction catalyzed by water and by oxalic acid at different altitudes.

[revised manuscript text omitted]

[a]The rate $v_{SO3\cdots H2O+OA}$ represents the sum of reaction rate for $SO_3\cdots H_2O$ complex with four oxalic acid conformers (cTt, tTt, tCt, cCt). The rate ($v_{SO3\cdots H2O+cTt}/v_{SO3\cdots H2O+H2O}$, $v_{SO3\cdots H2O+tTt}/v_{SO3\cdots H2O+H2O}$, $v_{SO3\cdots H2O+tCt}/v_{SO3\cdots H2O+H2O}$, $v_{SO3\cdots H2O+cCt}/v_{SO3\cdots H2O+H2O}$) are shown in Table S10.

The new Table 5:

**Table 5.** Relative rate of $SO_3$ hydration reaction catalyzed by oxalic acid and by water at different altitudes.

| altitude (km) | 0 | 0 | 2 | 4 | 6 | 8 | 10 | 12 |
|---|---|---|---|---|---|---|---|---|
| P (bar) | 1.01325 | 1.01325 | 0.795 | 0.617 | 0.472 | 0.357 | 0.265 | 0.194 |
| T (K) | 298.15 | 288.15 | 275.15 | 262.17 | 249.19 | 236.22 | 223.25 | 216.65 |
| $^a v_{OA1}/v_{w1}$ | $3.87\times10^{-6}$ | $1.43\times10^{-5}$ | $3.47\times10^{-5}$ | $1.20\times10^{-4}$ | $4.43\times10^{-4}$ | $1.95\times10^{-3}$ | $1.80\times10^{-2}$ | $9.53\times10^{-2}$ |

[a]The rate $v_{OA1}$ represents the sum of reaction rate for Reaction X1 (X=cTt, tTt, tCt, cCt). The rate ($v_{cTt1}/v_{w1}$, $v_{tTt1}/v_{w1}$, $v_{tCt1}/v_{w1}$, $v_{cCt1}/v_{w1}$) are shown in Table S10.

As these data have been changed, the corresponding discussion is also different. We rewrote these corresponding contents in the revised manuscript. These new contents also are shown as follows:

Page 10, line 13 - 16: As shown in Table 1, the rate constant of the Reaction 1 changes from $4.21 \times 10^{-31}$ $cm^6$ $molecule^{-2}$ $s^{-1}$ (298.15 K at 0 km altitude) to $3.92 \times 10^{-27}$ $cm^6$ $molecule^{-2}$ $s^{-1}$ (216.15 K at 12 km altitude). For the Reaction X1, the transformation of rate constants in the range of altitudes can also be found ($k_{cTt1}$: from $6.45 \times 10^{-28}$ to $3.35 \times 10^{-22}$; $k_{tTt1}$: from $2.32 \times 10^{-26}$ to $4.97 \times 10^{-20}$; $k_{tCt1}$: from 4.08

$\times 10^{-26}$ to $1.35 \times 10^{-19}$; $k_{cCt1}$: from $1.39 \times 10^{-25}$ to $6.61 \times 10^{-19}$ $cm^6$ $molecule^{-2}$ $s^{-1}$).

Page 10, line 17 - 19: Obviously, the rate constant in the oxalic acid-catalyzed $SO_3$ hydration reaction is about $10^3$ - $10^8$ times larger than that for water-catalyzed $SO_3$ hydration reaction within the range of altitudes.

Page 11, line 20 - 23: As shown in Table 5, reaction rate ratios between the $SO_3$ hydration reactions catalyzed oxalic acid conformers (cTt, tTt, tCt and cCt) and the $SO_3$ hydration reaction catalyzed by $H_2O$ are described. At an altitude of 0 km, the rate ratio for these two reactions is in the range of $10^{-5}$ - $10^{-6}$ at two temperatures (298.15 K and 288.15 K), which indicates that the oxalic acid-catalyzed $SO_3$ hydration is of minor importance at 0 km with different temperatures.

Page 12, line 1 - 5: When the altitude increases to 10 km, the oxalic acid-catalyzed reaction is just 2 orders of magnitude slower than water-catalyzed reaction. At 12 km altitude, the water-catalyzed hydration reaction is approximately 10 times faster than the oxalic acid-catalyzed $SO_3$ hydration. To sum up, the comparison of relative rate shows that, in the upper troposphere, the oxalic acid can play a significant role in enhancing $SO_3$ hydration to $H_2SO_4$.

7. The authors mentioned that "hydroperoxy radical, formic acid, sulfuric acid (TorrentSucarrat et al., 2012), nitric acid and ammonia have been reported to replace the second water to catalyze the hydration reaction of $SO_3$." I think the authors should compare the catalytic effect of these species with oxalic acid?

Reply: Thank you for your advice. We have compared the catalytic effect of these species with oxalic acid, and found that the oxalic acid has a good performance. That is, the catalytic effect of oxalic acid in $SO_3$ hydration reaction is better than nitric acid, sulfuric acid, and similar to formic acid, but weaker than ammonia. The added content are shown in revised manuscript (Page 10, line 21 - 23 and Page 11, line 1 - 7), which also is put below:

As seen from Table2, at 298.15 K, the oxalic acid-catalyzed $SO_3$ hydration reaction is $10^3$ - $10^5$ faster than the corresponding water-catalyzed reaction ($k_{cTt1}/k_{w1}$: $1.53 \times 10^3$; $k_{tTt1}/k_{w1}$: $5.50 \times 10^4$; $k_{tCt1}/k_{w1}$: $9.70 \times 10^4$; $k_{cCt1}/k_{w1}$: $3.31 \times 10^5$). In order to evaluate the catalytic effect of oxalic acid, we also list the rate constant ratio between other species and water catalyzed $SO_3$ hydration reaction. At 298

K, the nitric acid catalyzed rate constant is just 1.19 times larger than water catalyzed rate constant (Long et al., 2013). When sulfuric acid functions as autocatalyst, the value of rate constant ratio is around $10^2$ (Torrent-Sucarrat et al., 2012). When the formic acid acts as a catalyst, its rate constant is $10^4$ greater than that of water-catalyzed $SO_3$ hydration at 300 K (Hazra and Sinha, 2011). The rate constant for ammonia catalyzed $SO_3$ hydration is 7 orders of magnitude larger than that for water catalyzed rate constant at 298 K (Bandyopadhyay et al., 2017). These results indicate that the catalytic effect of oxalic acid in $SO_3$ hydration reaction is better than nitric acid, sulfuric acid, and similar to formic acid, but weaker than ammonia.

8. The paper is required to be revised by native English speakers.

**Reply:** Thank you for your advice. The English language is edited by an English teacher. The revision has been marked in green color in the revised manuscript. We also put these revisions below:

Page 1, line 13: "Some atmospheric species can involve in and facilitate the reaction" is modified to "Some atmospheric species can be involved in and facilitate the reaction".

Page 1, line 15: "The energy barrier of $SO_3$ hydration reaction catalyzed by oxalic acid (cTt, tTt, tCt and cCt conformers) is about or below 1 kcal mol$^{-1}$, ..." is modified to " The energy barrier of $SO_3$ hydration reaction catalyzed by oxalic acid (cTt, tTt, tCt and cCt conformers) is a little higher or less than 1 kcal mol$^{-1}$, ...".

Page 2, line 4: "These atmospheric HAT reactions have a main feature that two-point hydrogen bond can occur ..." is modified to " These atmospheric HAT reactions display a main feature that two-point hydrogen bond can occur ...".

Page 2, line 8: "Thus, the effect of catalysts on promoting atmospheric HAT reactions has attracted more attention of atmospheric scientists" is modified to "Thus, the effect of catalysts on promoting atmospheric HAT reactions has attracted more attention from atmospheric scientists".

Page 2, line 11: "For the reaction $SO_3 + H_2O \rightarrow H_2SO_4$, the pre-reactive $SO_3 \cdots H_2O$ complex firstly is formed, and the complex then rearrange to form $H_2SO_4$, which was proposed by Castleman et al" is modified to "For the reaction $SO_3 + H_2O \rightarrow H_2SO_4$, the pre-reactive $SO_3 \cdots H_2O$ complex is formed firstly, and the complex is then rearranged to produce $H_2SO_4$, which was proposed by Castleman et al".

Page 2, line 19: "It has also been shown that some other atmosphere molecules can behave as a catalyst to promote the hydration of $SO_3$" is modified to "It has also been shown that some other atmospheric molecules can serve as a catalyst to promote the hydration of $SO_3$".

Page 3, line 2: "In addition to accumulating in aerosols ..." is modified to "In addition to its accumulation in aerosols ...".

Page 3, line 6: "... that it can form stable complexes with water ..." is modified to "... that it can generate stable complexes with water ...".

Page 3, line 8: "The potential of oxalic acid for contributing to the NPF is mainly attributed to its capability of forming hydrogen bond with hydroxyl and/or carbonyl-type functional group" is modified to "For oxalic acid, its potential to promote the NPF is mainly attributed to its capability of forming hydrogen bond with hydroxyl and/or carbonyl-type functional group".

Page 3, line 14: "As is known, oxalic acid can exist in several conformational forms ..." is modified to "It is known that oxalic acid can exist in several conformational forms ...".

Page 3, line 22: "The geometric structures including all reactant, complex, transition state and products were optimized using M06-2X method (Zhao and Truhlar, 2008) with 6-311++G(3df,3pd) basis set" is modified to "The geometric structures, including all reactants, complexes, transition states and products, were optimized using M06-2X method (Zhao and Truhlar, 2008) with 6-311++G(3df,3pd) basis set".

Page 4, line 2 - 3:"... through the criterion that no imaginary frequencies for the local minimum point and one imaginary frequency for transition states" is modified to "... through the criterion that there are

no imaginary frequencies for the local minimum point and one imaginary frequency for transition states".

Page 4, line 8: "To obtain the conformational population of oxalic acid in different temperature more accurately ..." is modified to "To obtain more accurate conformational population of oxalic acid in different temperature ...".

Page 4, line 16 - 17: "On the basis of the discussion in this paper, it can conclude that the hydration reactions begin with the formation of pre-reactive complex, and then undergo a transition state to form post-reactive complex" is modified to "Base on the discussion in this paper, it can be concluded that $SO_3$ hydration reactions begin with the formation of pre-reactive complex, and then pass by a transition state to form a post-reactive complex".

Page 5, line 2: "... $k_{uni}$ is the rate constant for unimolecular reaction of pre-reactive complex to post-reactive complex" is modified to "... $k_{uni}$ is the rate constant for a unimolecular reaction from the pre-reactive complex to post-reactive complex".

Page 5, line 10 - 11: "For oxalic acid-catalyzed hydration reaction of $SO_3$, it also has two reaction channels and has the similar features as the water-assisted hydration process" is modified to "For oxalic acid-catalyzed hydration reaction of $SO_3$, there are two reaction channels and the same features as in the water-assisted hydration process".

Page 6, line 16 - 17: "Although the hydration of $SO_3$ involving two water molecules has been talked about many times, we still include it in our paper so as to compare this reaction with ..." is modified to "Although the hydration of $SO_3$ involving two water molecules has been discussed many times, we still include it in our paper so as to compare it with ...".

Page 7, line 2 - 4: "The one is that water dimer react with $SO_3$ to obtain pre-reactive complex, then this complex rearrange to form $H_2SO_4 \cdots H_2O$ complex (channel 1); the other begin with the reaction of $SO_3 \cdots H_2O$ complex with water, the following rearrangement is the same as the channel 1 (channel 2)"

is modified to "One is that water dimer reacts with $SO_3$ to obtain the pre-reactive complex, then this complex is rearranged to form $H_2SO_4\cdots H_2O$ complex (channel 1); the other begins with the reaction of $SO_3\cdots H_2O$ complex with water, the following reaction process is the same as the channel 1 (channel 2)".

Page 8, line 6: " The failure of this transfer is due to that the hydrogen and oxygen atom involving in two-point hydrogen bond do not come from the same carboxyl group" is modified to "The transfer failure is attributed to the fact that the hydrogen and oxygen atom involving in two-point hydrogen bond do not come from the same carboxyl group".

Page 9, line 1 - 5: " In the two channels starting from the $tCt\cdots H_2O + SO_3$ entry and $SO_3\cdots H_2O + tCt$ entry, the same pre-reactive complex ($RC_{tCt}$) can be formed with the binding energy of 13.11 kcal mol$^{-1}$, 15.56 kcal mol$^{-1}$, respectively, with respect to the two reactants. The $RC_{tCt}$ proceeds via the transition state ($TS_{tCt}$) lying above $RC_{tCt}$ by 0.28 kcal mol$^{-1}$ into post-reactive complex ($PC_{tCt}$), which is 14.01 kcal mol$^{-1}$ more stable than the $RC_{tCt}$ complex. The $PC_{tCt}$ complex also can be formed from the $H_2SO_4$ and tTt conformer with the energy release of 17.73 kcal mol$^{-1}$" is modified to "In the two channels, the same pre-reactive complex ($RC_{tCt}$) can be formed with the binding energy of 12.92 kcal mol$^{-1}$ relative to $tCt\cdots H_2O + SO_3$, 15.63 kcal mol$^{-1}$ with respect to $SO_3\cdots H_2O + tCt$, respectively. The $RC_{tCt}$ proceeds via the transition state ($TS_{tCt}$) (with energy barrier of 0.05 kcal mol$^{-1}$) into post-reactive complex ($PC_{tCt}$). The $PC_{tCt}$ complex can also be generated from the $H_2SO_4$ and tTt conformer releasing 16.98 kcal mol$^{-1}$ of energy".

Page 9, line 14: "Another thing we want to mention is that ..." is modified to "Another point worth mentioning is that ...".

Page 11, line 8: " Based on the calculated Gibbs free energy at G4 level (see Table S8), and assuming a Boltzmann distribution, the mole fractions for oxalic acid conformers can be obtained (Table 3)" is modified to "Based on the calculated Gibbs free energy at G4 level (see Table S8) and an assumption of Boltzmann distribution, mole fractions for oxalic acid conformers can be obtained (Table 3)".

Page 12, line 21: "The results signify oxalic acid has the higher catalytic ability than water for $SO_3$ hydration to form $H_2SO_4$" is modified to "The results demonstrate that oxalic acid has the stronger catalytic ability than water for $SO_3$ hydration to form $H_2SO_4$".